# Changes to the chemical state of the northern hemisphere atmosphere during the second half of the twentieth century

**Mike J. Newland[1],\* Patricia Martinerie[2], Emmanuel Witrant[3], Detlev Helmig[4], David R. Worton[5], Chris Hogan[1], William T. Sturges[1], Claire E. Reeves[1]**

[1] {Centre for Ocean and Atmospheric Sciences, School of Environmental Sciences, University of East Anglia, Norwich, UK}

[*] {now at: Wolfson Atmospheric Chemistry Laboratories, Department of Chemistry, University of York, York, UK}

[2] {Univ. Grenoble Alpes/CNRS, LGGE, F-38000 Grenoble, France}

[3] {Univ. Grenoble Alpes/CNRS, GIPSA-Lab, F-38000 Grenoble, France}

[4] {Institute of Arctic and Alpine Research, University of Colorado, Boulder, Colorado, USA}

[5] {National Physical Laboratory, Teddington, Middlesex, UK}

Correspondence to: M. J. Newland (mike.newland@york.ac.uk)

**Abstract**

**The $NO_X$ (NO and $NO_2$) and $HO_X$ (OH and $HO_2$) budgets of the atmosphere exert a major influence on atmospheric composition, controlling removal of primary pollutants and formation of a wide range of secondary products, including ozone, that can influence human health and climate. However, there remain large uncertainties in the changes to these budgets over recent decades. Due to their short atmospheric lifetimes, $NO_X$ and $HO_X$ are highly variable in space and time, and so the measurements of these species are of limited value for examining long term, large scale changes to their budgets. Here, we take an alternative approach by examining long-term atmospheric trends of alkyl nitrates, secondary oxidation products of alkanes, the production efficiency of which is dependent on the atmospheric [NO]/[$HO_2$] ratio. We derive long term trends of three alkyl nitrates (2-butyl nitrate, 2+3-pentyl nitrate, 3-methyl-2-butyl nitrate) from measurements in firn air from the NEEM site, Greenland. Their mixing ratios increased by a factor of 3 – 4 between the 1970s and 1990s. This was followed by a steep decline to the sampling date of**

**2008. We then examine how the trends in the alkyl nitrates compare to similarly derived trends in their parent alkanes. The ratios of the alkyl nitrates to their parent alkanes increase from around 1970 to the late 1990's. This is consistent with large changes to the [NO]/[HO$_2$] ratio in the northern hemisphere atmosphere during this period. Alternatively, they could represent changes to concentrations of the hydroxyl radical, OH, or to the transport time of the air masses from source regions to the Arctic.**

## 1   Introduction

The NO$_X$ (NO + NO$_2$) and HO$_X$ (OH + HO$_2$) budgets of the troposphere act to control the concentrations of oxidants such as OH, ozone and NO$_3$ (Fig. 1). These in turn control removal of pollutants from the atmosphere. Emissions of NO$_x$ in the northern hemisphere are mainly anthropogenic, with roughly equal proportions from power generation and transport (Olivier and Berdowski et al., 2001; Olivier et al., 2001). NO$_X$ and HO$_X$ are linked through ozone production, which is positively correlated with NO$_X$ concentrations in the background atmosphere through the photolysis of NO$_2$ (Reactions R1-R2). The photolysis of ozone in the presence of water vapour then leads to the production of OH (Reactions R3-R4). Other processes, such as alkene ozonolysis (Johnson and Marston, 2008) and photolysis of HONO (formed from heterogeneous reactions of NO$_2$ (Stone et al. 2012)) may also be important primary sources of HO$_X$, particularly in winter (e.g. Heard et al., 2004).

Removal of NO$_X$ from the atmosphere is controlled by the reaction of NO$_2$ with OH during the daytime (Reaction R5). This forms nitric acid, HNO$_3$, which is lost from the atmosphere by wet deposition. At night, and during the winter, the heterogeneous reaction of the NOx reservoir species N$_2$O$_5$ (formed from the reaction of NO$_2$ with NO$_3$ (Reactions R6-R7) with H$_2$O on aerosol becomes an important NO$_X$ sink (Reaction R8). OH and HO$_2$ rapidly interconvert through the reactions of OH with CO and hydrocarbons, such as alkanes, and the reaction of HO$_2$ with NO (Reaction R9). The reaction of NO with peroxy radicals (HO$_2$ and RO$_2$ – Reactions R9-R10) recycles the NO back to NO$_2$. The main removal process for HOx in urban regions is the reaction of OH with NO$_2$ (Reaction R5) (Stone et al., 2012), while HO$_2$ self-reaction and reaction with RO$_2$ (in particular CH$_3$O$_2$) (Reactions R11-R12) dominate in low NOx environments (Stone et al., 2012).

***HO_X sources***
$$NO_2 \xrightarrow{h\nu} NO + O \hspace{4cm} \text{(R1)}$$
$$O + O_2 + M \rightarrow O_3 + M \hspace{3.5cm} \text{(R2)}$$
$$O_3 \xrightarrow{h\nu} O(^1D) + O_2 \hspace{4cm} \text{(R3)}$$
$$O(^1D) + H_2O \rightarrow 2OH \hspace{3.8cm} \text{(R4)}$$
***NO_X sinks***
*Day:*    $$NO_2 + OH \rightarrow HNO_3 \hspace{3.5cm} \text{(R5)}$$
*Night:*    $$NO_2 + O_3 \rightarrow NO_3 + O_2 \hspace{3cm} \text{(R6)}$$
$$NO_2 + NO_3 \leftrightharpoons N_2O_5 \hspace{3.5cm} \text{(R7)}$$
$$N_2O_5 + H_2O \xrightarrow{het.} 2HNO_3 \hspace{3cm} \text{(R8)}$$
***NO_X and HO_X recycling***
$$NO + HO_2 \rightarrow OH + NO_2 \hspace{3cm} \text{(R9)}$$
$$NO + RO_2 \rightarrow RO + NO_2 \hspace{3cm} \text{(R10)}$$
***HO_X sinks***
$$HO_2 + RO_2 \rightarrow ROOH \hspace{3.3cm} \text{(R11)}$$
$$HO_2 + HO_2 \rightarrow H_2O_2 + O_2 \hspace{2.8cm} \text{(R12)}$$

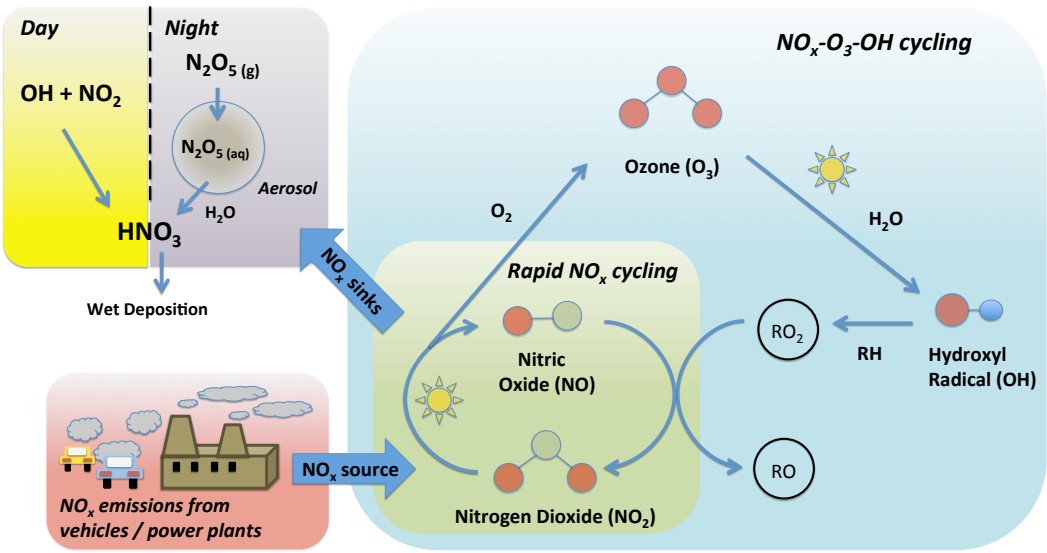

Figure 1 Schematic of the $NO_x$-$O_3$-OH relationship in the background troposphere.
However, changes to the atmospheric concentrations of both $HO_X$ and $NO_X$ during the previous
century are poorly constrained. This is because all $HO_X$ and $NO_X$ species are short lived, present
at low concentrations (0.01 – 10 ppt), and have a high spatial and temporal variability (e.g.
Stone et al., 2012). This makes them difficult to measure and trends difficult to identify (based
on spatially and temporally variable data sets). Furthermore, a range of state of the art
atmospheric chemistry transport models give no consensus of even the sign of OH change
between 1850 and 2000 (Naik et al., 2013). However, the models do agree that between 1980
and 2000 there has been an increase in northern hemisphere OH concentrations, with the best
estimate of the increase being 4.6 (± 1.9) %. This modelled increase is driven by increases in
the $NO_X$ burden and in the water vapour concentration.
To attempt to study historical trends in $HO_X$ and $NO_X$ we have examined trends in longer lived
species which are affected by changes to $HO_X$ and $NO_X$ in the atmosphere.
In this paper we report long term atmospheric trends of three alkyl nitrates (2-butyl nitrate, 2+3-
pentyl nitrate, 3-methyl-2-butyl nitrate) derived from Arctic firn air. These are chemically
produced in the atmosphere from the oxidation of alkanes and subsequent reaction of the peroxy
radical formed with NO. The alkyl nitrate records are combined with previously reported trends
of their parent alkanes from the same Arctic firn site. These records provide a proxy from which
we can learn about the chemical state of the atmosphere at the time they were formed.

## 1.1 Alkanes

Emissions of butanes ($C_4H_{10}$) and pentanes ($C_5H_{12}$) to the atmosphere are almost entirely anthropogenic ($> 98$ % globally (Pozzer et al., 2010)), associated with fugitive emissions during oil and natural gas extraction and transmission, and evaporation and combustion of fossil fuels, such as in road vehicles (Pozzer et al., 2010; Pétron et al., 2012; Helmig et al., 2014a). Butane and pentane emissions from vehicles will be dependent on fuel composition, with evaporative emissions also dependent on temperature. Many areas in North America are part of 'ozone attainment areas', and during summer months (June – September 15) have been required by law since 1990 to provide a gasoline blend with a low Reid vapour pressure (RVP) to reduce the ozone production potential (www.epa.gov). This reduction in RVP is generally achieved by reducing the fuel's butane content relative to winter-time fuel (e.g. Gentner et al., 2009).

Measurements in firn air from Greenland (Aydin et al., 2011; Worton et al., 2012; Helmig et al., 2014) suggest northern hemisphere C2-C5 alkane mixing ratios increased through the middle of the past century to a peak in ~1980 (~1970 for ethane) and then declined to roughly 1960 levels by 2000. In-situ measurements from the urban areas of London (1993 – 2008) (Dollard et al., 2007; von Schneidemesser et al., 2010) and Los Angeles (1960 – 2010) (Warneke et al., 2012) show steadily decreasing alkane mixing ratios, as do measurements at the semi-rural site of Hohenpeissenberg, Germany (von Schneidemesser et al., 2010). Emission estimates from the ACCMIP global emission inventory (Lamarque et al., 2010) (available at http://eccad.sedoo.fr) show butane and pentane emissions in Europe and North America increasing steadily between 1950 and 1980 before falling again to roughly 1965 levels by 2000.

The primary removal mechanism of alkanes from the atmosphere is reaction with the hydroxyl radical, OH (minor sinks include reaction with atomic chlorine, Cl, and the nitrate radical, $NO_3$).

Atmospheric mixing ratios of butanes and pentanes display a large seasonal cycle in mid-high latitudes (*e.g.* Swanson *et al.*, 2003; Helmig *et al.*, 2009) due to changes in their chemical lifetimes (~1 month in the winter and 4 – 5 days in the summer) driven by the seasonal cycle in OH concentration.

## 1.2 Alkyl Nitrates

Alkyl nitrates ($RONO_2$) are secondary oxidation products of alkanes (RH). Their atmospheric lifetimes are on the order of months in winter and ten days in summer (Clemitshaw et al., 1997). Consequently, they display a strong seasonal cycle in the Arctic, with peaks in the late

winter/early spring and minima in the summer (Swanson et al., 2003), similar to the alkanes.
Alkyl nitrates are formed when alkanes react with OH to form a peroxy radical, $RO_2$ (Reaction
R13), which subsequently reacts with NO to form an alkyl nitrate (Reaction 14b) (*e.g.* Talukdar
et al., 1997). This is a minor channel of the $RO_2 + NO$ reaction (Reaction R14a) which generally
leads to ozone production via recycling of NO to $NO_2$ and the subsequent photolysis of $NO_2$.
$RO_2$ can also react with $HO_2$ (the hydroperoxyl radical) (Reaction R11) to form a peroxide
(ROOH). The probability of $RO_2$ reacting with NO (leading to alkyl nitrate production) is thus
governed by the ratio $[NO]/[HO_2]$. Alkyl nitrates are lost from the atmosphere by reaction with
OH (Reaction R15), photolysis (Reaction R16) and wet/dry deposition.
$$RH + OH \xrightarrow{O_2} RO_2 + H_2O \qquad\qquad k_{13}, \alpha_{13} \qquad\qquad (R13)$$
$$RO_2 + NO \longrightarrow RO + NO_2 \qquad\qquad k_{14}, (1-\alpha_{14}) \qquad\qquad (R14a)$$
$$\longrightarrow RONO_2 \qquad\qquad k_{14}, \alpha_{14} \qquad\qquad (R14b)$$
$$RO_2 + HO_2 \longrightarrow ROOH \qquad\qquad k_{11} \qquad\qquad (R11)$$
$$RONO_2 + OH \longrightarrow products \qquad\qquad k_{15} \qquad\qquad (R15)$$
$$RONO_2 \xrightarrow{hv} products \qquad\qquad j_{16} \qquad\qquad (R16)$$
## 2 Methodologies
### 2.1 Firn Sampling
Firn air samples were collected at the NEEM site, Greenland (77.45°N, 51.07°W, 2484m a.s.l)
from two boreholes between 14[th] and 30[th] July 2008 ("2008 EU hole" and "2008 US hole").
Further samples were collected from another NEEM borehole during July 2009 ("2009 hole").
The '2008 EU hole' was sampled using the firn air system of the University of Bern (Schwander
et al., 1993), and the 'US' hole, sampled using the US firn air system (Battle et al., 1996). The
alkane measurements used in this work – originally reported in Helmig et al. (2014b) - come
from a combination of the 2008 EU and US holes and the 2009 hole with the exception of the
pentanes, which come only from the 2008 EU and US holes. The alkyl nitrate samples come
only from the 2008 EU hole. Full sampling details are available in Helmig et al. (2014b) and
Buizert et al. (2012).

## 2.2 Firn Analysis

The firn air samples from the 'EU' hole at NEEM were analysed for alkyl nitrates at UEA using a GC-MS in Negative Ion Chemical Ionisation mode (GC-NICI-MS) (e.g. Worton et al., 2008).

2-pentyl nitrate and 3-pentyl nitrate are presented together as 2+3-pentyl nitrate because the two are not baseline separated in the chromatogram.

The NEEM samples were analysed using the UEA calibration scale. This was converted to the NCAR scale (against which the North GRIP 2-butyl nitrate and 2+3-pentyl nitrate samples are calibrated) for direct comparison with the North GRIP atmospheric histories from Worton et al. (2012) and with the in-situ measurements at Summit by UCI (Swanson et al., 2003; Dibb et al., 2007). This scaling was based on an inter-comparison between the UEA and NCAR standards in 2005 and 2012/13. These led to a rescaling of the UEA 2-butyl nitrate values by 1.245 and 2+3-pentyl nitrate by 1.409. The measurements of 3-methyl-2-butyl nitrate were not rescaled as the North GRIP measurements were made on the UEA scale.

Firn air samples from the 'EU' hole at NEEM were analysed for alkanes at the Max Planck Institute Laboratory (MPI) by gas chromatography with flame ionisation detection (GC-FID) (see Baker et al., 2010 for further details). At the Institute of Arctic and Alpine Research (INSTAAR) firn air samples were analysed from both the 'EU' and 'US' holes at NEEM for alkanes by GC-FID (see Pollmann et al., 2008 and Helmig et al., 2014b for further details).

## 2.3 Firn Modelling

The air sampled from any given depth in the firn column is representative of a range of ages because of the inter-connected nature of the firn. Firn models can be used to derive the atmospheric history of a gas from measurements of air trapped in the firn. The extent and rate, at which the gas diffuses through the firn, depends on the diffusivity profile of the firn, the diffusivity coefficient of the gas, and on the gravitational fractionation (caused by the molecular weight) of the gas. The diffusion profile is different for every firn site.

For determining the atmospheric history of a gas from firn air measurements, the firn diffusion profiles must first be constrained. This is done using a series of reference gases with well known atmospheric histories. At NEEM the reference gases used were $CO_2$, $CH_4$, $SF_6$, HFC-134a, CFC-11, CFC-12, CFC-113, and $CH_3CCl_3$, as well as $^{14}CO_2$ (Witrant et al., 2012).

Each gas also has a different diffusion rate through the firn based on its molecular structure,
this is called the diffusion coefficient. The diffusion coefficient is calculated relative to a
reference gas, generally $CO_2$. Different methods have been reported for the calculation of these
diffusion coefficients (e.g. Chen and Othmer, 1962; Fuller et al., 1966). The diffusion
coefficients of the alkyl nitrates were calculated using the method of Fuller et al. (1966) based
on the sum of the Le Bas molar volumes of the molecule. Model runs were also performed
using diffusion coefficients for the alkyl nitrates calculated using the Chen and Othmer method.
These coefficients are ~ 10% lower than those calculated using the Fuller method. However,
the atmospheric scenarios derived from the modelling are very similar, well within the 2-σ
uncertainty envelopes presented in Figure 2. The diffusion coefficients used for the firn
modelling for each molecule within this work are given in Table 1.
The inverse model used for the atmospheric history reconstructions was the most recent version
of the LGGE-GIPSA atmospheric trend reconstruction model described in Witrant and
Martinerie (2013).
The atmospheric mole fraction derived from the firn reconstructions represents an annual mean.
The alkanes and alkyl nitrates examined in this work have a strong atmospheric seasonality due
to changes in their chemical lifetimes driven by seasonal variability in OH concentration in the
air masses in which they are transported to the Arctic. Thus the seasonal cycle of both species
follows a roughly sinusoidal curve with a peak in the late winter (March) and a trough in mid-
summer (July-August) (Swanson et al. 2003). Consequently, changes to the firn derived mole
fractions are likely to be dominated by changes to winter-time atmospheric concentrations.
This model cannot take into account the seasonality in the signal that is preserved in the upper
part of a firn profile. Therefore, measurements above a certain depth must be excluded from the
model input. It is noted that the latter part (post-1995) of the model derived scenarios for 2+3-
pentyl nitrate is rather sensitive to the inclusion or exclusion of the measurement at 34.72 m
(the shallowest measurement used). The scenarios presented in this work are based on including
this measurement.

Table 1 Diffusion coefficients used in the firn modelling, calculated from Le Bas molecular volumes using the method of Fuller *et al.* (1966).

| Compound | Diffusion Coefficient relative to $CO_2$ |
| --- | --- |
| 2-butyl nitrate | 0.467 |
| 2+3-pentyl nitrates | 0.428 |
| 3-methyl-2-butyl nitrate | 0.428 |

## 3 Alkyl Nitrate Trends

Atmospheric histories of the three alkyl nitrates 2-butyl nitrate, 2+3-pentyl nitrate, and 3-methyl-2-butyl nitrate (formed from n-butane, n-pentane, and iso-pentane respectively) derived from firn air measurements at NEEM are shown in Figure 2. The records of all three alkyl nitrates show similar features (as would be expected from the similar sources and sinks). All show a steep increase in mixing ratio from the 1970s to the 1990s with increases of a factor of 3 − 4. The peak in the 1990s is followed by a steep decline to the sampling date of 2008.

Figure 2 also shows the atmospheric histories of the same three alkyl nitrates derived from firn air from North GRIP, Greenland, up to 2001, presented in Worton et al. (2012) (pink shaded area). There is very good agreement between the derived trends at the two sites. Differences can be attributed to the limited number of measurements at both sites, possible drift in the calibration standard used, and uncertainties in the firn modelling. Both sites show the same large increase in mixing ratios from the 1970s to the 1990s. Importantly, the NEEM records show that the turnover and subsequent decline in mixing ratios, the beginnings of which were evident in the North GRIP records, appears to continue through the 2000s. However, as noted in Section 2.3, the derived atmospheric history of 2+3-pentyl nitrate is sensitive to the inclusion of the measurement at 34.72 m. A scenario that did not include this measurement was almost flat from 1995 to 2008 rather than declining as in Figure 2.

There are very limited in-situ measurements of alkyl nitrates in the Arctic and even fewer that cover a whole seasonal cycle. Swanson et al. (2003) report the seasonal cycle of 2-butyl nitrate at the Summit station, Greenland (72.34 N, 38.29 W, 3250 m a.s.l), from June 1997 to June 1998 based on samples collected roughly every two days. Dibb et al. (2007) report monthly mean measurements of 2-butyl nitrate for the period June 2000 through to August 2002 based on samples taken roughly weekly also from Summit. In order to compare these in-situ

measurements to output derived from the firn measurements the annual mean is taken. This is
because the firn smoothes out the seasonality and represents the annual mean of mixing ratios.
Calculating the 2-butyl nitrate annual mean for the three periods 1997-1998 (6.8 ppt), 2000-
2001 (5.3 ppt), and 2001-2002 (5.0 ppt) gives values that can be compared to the output from
the firn model for 2-butyl nitrate. These agree with the firn model output in terms of absolute
mixing ratios of 2-butyl nitrate during this period (5 – 8 ppt) (Fig. 2). They also show a declining
trend through the period, in agreement with the firn model output, though this is not statistically
significant within the uncertainties.

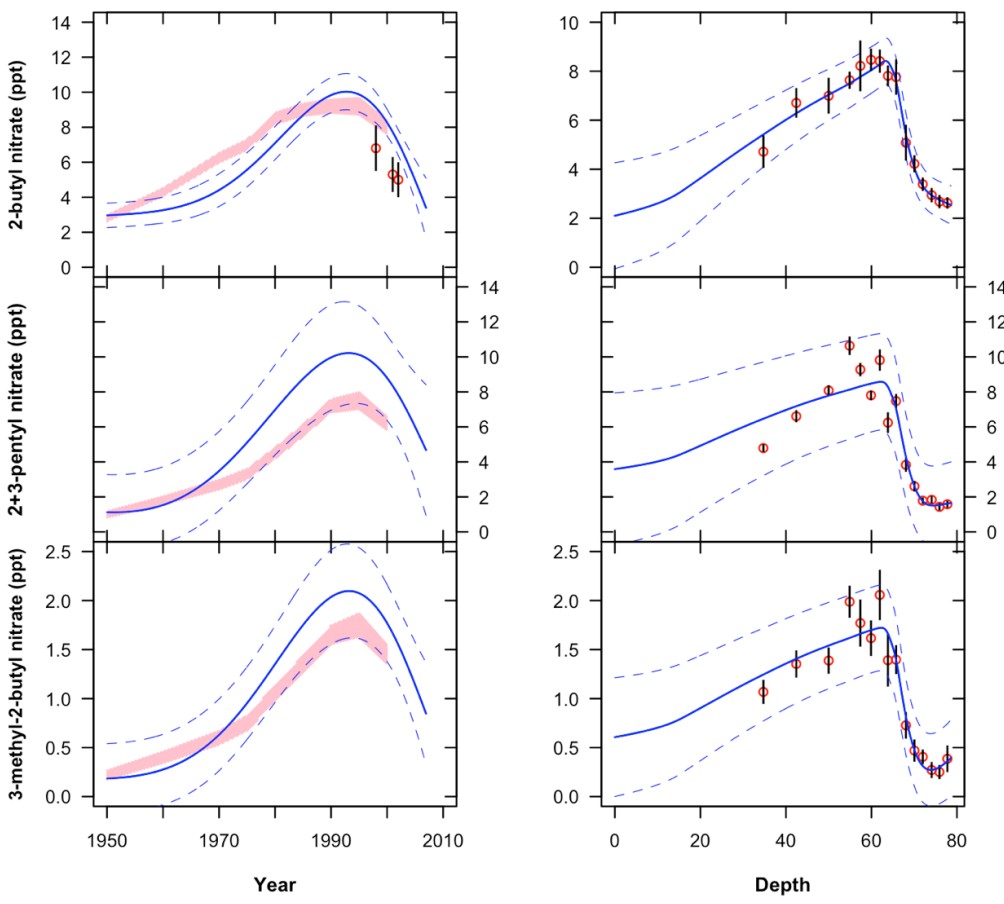

Figure 2 Concentration-depth profiles in the firn and the model derived atmospheric histories. Right panel: The
concentration-depth profiles measured in the firn (ppt): red open circles: measured mixing ratios (ppt) with error
bars indicating the 2-σ uncertainty; solid blue line: best fit of the firn model, dashed blue lines indicate the 2-σ
combined analytical and model uncertainties. Left panel: Atmospheric histories of the alkyl nitrates derived from
the firn air measurements using the inverse modelling technique described within (solid blue lines). Dashed lines
represent the 2-σ confidence margins of the model calculations, combining the analytical and model uncertainties.
Pink shaded area: atmospheric histories presented in Worton et al. (2012) derived from firn air measurements at
North GRIP, Greenland. Red open circles: Annual average of in-situ measurements at Summit, Greenland (see
text for details) with 1-σ uncertainty.

Considering Reactions R13 – R16, the trends in the alkyl nitrate mixing ratios (Fig. 2) could be caused by:

(i) Changes to the atmospheric mixing ratios of the parent alkanes;

(ii) Changes to $[OH]t$, where $t$ is time since emission of the alkane. i.e. the amount of photochemical processing that the air mass in which the alkyl nitrates are being formed undergoes before reaching the Arctic;

(iii) Changes to the production efficiency of the alkyl nitrates, i.e. whether the peroxy radical reacts with NO (Reaction R14) or with $HO_2$ (Reaction R11);

(iv) Changes to the alkyl nitrate sinks, i.e., changes in [OH] or radiation.

Concerning point (i), the peak in alkyl nitrate mixing ratios in the 1990s is not contemporaneous with that of the parent alkanes (~1980 – Figure 3). This suggests that the changes to the alkyl nitrate mixing ratios are not being primarily driven by changes to the parent alkane. By considering the ratio of the alkyl nitrate to its parent hydrocarbon, using the firn derived alkane trends from NEEM presented in Helmig et al. (2014b), we can effectively remove the effect of changes to the parent hydrocarbon from the alkyl nitrate signal. This is done in Section 4.

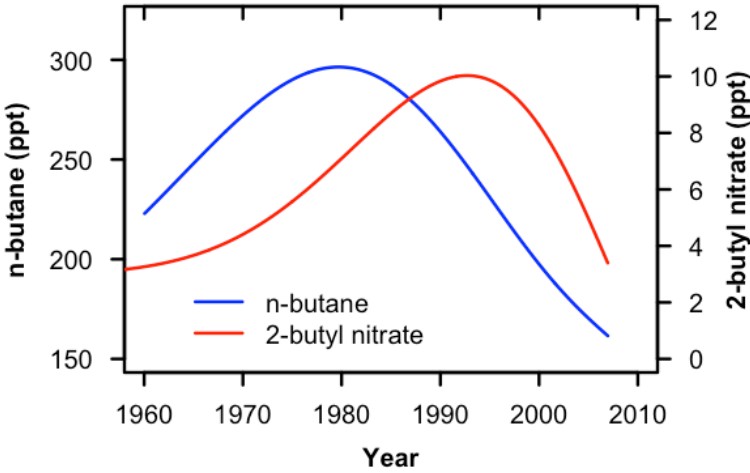

Figure 3 Atmospheric histories of 2-butyl nitrate (red) and its parent alkane, n-butane (blue), derived from firn measurements at NEEM, Greenland.

Concerning point (iv), there is evidence for global dimming (i.e. a decrease in surface solar radiation) of about 5 % between 1960 and 1990 in the northern hemisphere. However, this

began to turn around during the mid 1980s and there was a brightening trend between 1985 and
2000 (Wild et al., 2005). This minor change to the alkyl nitrate sink is unlikely to have had a
noticeable effect on mixing ratios.
Points (ii), (iii), and (iv) are discussed further in the following sections.

## 6   4   Ratios of Alkyl Nitrate to Parent Alkane

Bertman et al. (1995) presented a mathematical equation to describe the production of alkyl
nitrates in a $NO_x$ rich environment (Equation E1 (assumes an initial zero mixing ratio for alkyl
nitrates)).

$$\frac{[RONO_2]}{[RH]} \quad = \quad \frac{\beta k_A}{(k_B - k_A)}\left(1 - e^{(k_A - k_B)t}\right) \qquad (E1)$$

Where $\beta = \alpha_{13}\alpha_{14}$, $k_A = k_{13}[OH]$, $k_B = k_{15}[OH] + j_{16}$ ; subscript numbers refer to reactions given
in the Introduction. In this equation, [OH] is assumed to be a constant. Similarly for the
purposes of this work, [OH] is assumed to represent an average [OH], $[\overline{OH}]$, to which the air
mass is exposed during transport from the source region to the Arctic, i.e. $1/t * \int[OH].dt$.
Bertman et al. (1995) derived Equation E1 by integrating the rate equation for $[RONO_2]$
assuming a $NO_X$ rich environment (Equation E2).

$$\frac{d[RONO_2]}{dt} = \beta k_A[RH] - k_B[RONO_2] \qquad (E2)$$

We extend Equation E2 to include the possibility of alkyl nitrate production at less than 100%
efficiency, in a non-NOx-rich environment, i.e. that the peroxy radical, $RO_2$, formed may react
with something other than NO. This is achieved by the inclusion of the term
$k_{14}[NO]/(k_{14}[NO]+\text{other } RO_2 \text{ sinks})$. In high-NOx environments, this value is $\cong 1$. However, in
lower NOx environments, other sinks for the peroxy radical, $RO_2$, will compete with NO. In
reality the term $k_{11}[HO_2]$ is likely to dominate the 'other $RO_2$ sinks' term in a background
environment. The only other species likely to be present at high enough concentrations to
compete with the $RO_2 + HO_2$ reaction is the methyl peroxy radical ($CH_3O_2$), which may be
present at similar concentrations to $HO_2$, but the reaction rate of $CH_3O_2$ with other alkyl peroxy
radicals larger than $CH_3O_2$ is $\leq 2\times10^{-13}$ $cm^{-3}$ $s^{-1}$ (IUPAC), two orders of magnitude slower than

1 the reaction with $HO_2$ (IUPAC). Hence in Equation E3 we extend Equation E2 by including the

2 term $k_{14}[NO]/(k_{14}[NO]+k_{11}[HO_2])$.

$$\frac{d[RONO_2]}{dt} = \frac{\beta k_A[RH]k_{14}[NO]}{k_{14}[NO]+k_{11}[HO_2]} - k_B[RONO_2] \tag{E3}$$

4 For the purposes of our calculations, $k_{14}[NO]/(k_{14}[NO]+k_{11}[HO_2])$ is assumed (in the same way

5 as [OH]) to represent an integrated value for this ratio during transport of the air mass from the

6 source region to the Arctic, i.e. $1/t * \int k_{14}[NO]/(k_{14}[NO]+k_{11}[HO_2]).dt$. We denote this term $\gamma$

7 (Equation E4).

$$\gamma = \overline{\left(\frac{k_{14}[NO]}{k_{14}[NO]+k_{11}[HO_2]}\right)} = Mean\ alkyl\ nitrate\ production\ efficiency \tag{E4}$$

9 $k_{14}[NO]/(k_{14}[NO]+k_{11}[HO_2])$ would not be expected to be constant in reality since [NO] is likely

10 to change by orders of magnitude during transport, with values on the order of $2.5\times10^{11}$ cm$^{-3}$

11 close to the emissions source, and falling to $\sim1\times10^{8}$ cm$^{-3}$ further from source. However, while

12 changes to the ratio $k_{14}[NO]/(k_{14}[NO]+k_{11}[HO_2])$ at different times along the air mass trajectory

13 will affect $d[RONO_2]/dt$ at that time differently because $d[RONO_2]/dt$ is also driven by [RH]

14 which is a function of time, the uncertainties introduced by the assumption of $\gamma$ as an integrated

15 value on $[RONO_2]/[RH]$ calculated at time $t = 10$ days are on the order of 5 % (see

16 Supplementary Information). The observed changes in $[RONO_2]/[RH]$ in the firn are

17 considerably larger than this, on the order of a factor of $3 - 5$. Hence we consider the assumption

18 of $\gamma$ as a constant to be a reasonable assumption for the sake of making the problem tractable

19 and that the changes to $\gamma$ that we calculate in the paper are not an artefact of this assumption.

20 Since $\gamma$ is treated as a constant, integration of Equation E3 gives an equation the same as

21 Equation E1 from Bertman et al. except with the addition of the term $\gamma$ (Equation E5).

$$\frac{[RONO_2]}{[RH]} = \frac{\gamma \beta k_A}{(k_B-k_A)}\left(1 - e^{(k_A-k_B)t}\right) \tag{E5}$$

23 Atmospheric histories of the three parent alkanes of the alkyl nitrates presented in Figure 2 were

24 presented in Helmig et al. (2014b) – Figure 7 (n-butane, n-pentane, iso-pentane). These are used

25 here, in conjunction with the alkyl nitrate histories in Figure 2, to determine trends of the ratio

26 $[RONO_2]/[RH]$ for each alkyl nitrate-alkane pair. By rearranging Equation E5, we can then

27 probe two of the possible causes for the observed alkyl nitrate trends. Firstly, that the mean

production efficiency (i.e. γ) has changed over the time period of the firn record. Secondly, that
the processing of the air mass, i.e. mean OH concentration, $[\overline{OH}]$, multiplied by the transport
time from source regions to the Arctic, $t$, has changed.

## 5    Changes to the Production Efficiency of the Alkyl Nitrates

In an urban environment, daytime [NO] can range from ten to a few hundred ppb. In this case,
the production efficiency of the alkyl nitrates ≈ 1, i.e. all of the alkyl peroxy radicals formed in
Reaction R13 go on to form alkyl nitrates at a yield determined by the branching ratio $\alpha_{13}$.
However, in rural and more remote regions of the atmosphere, daytime [NO] ranges from 1 –
100 ppt. At these mixing ratios $k_{14}[NO]/(k_{14}[NO]+k_{11}[HO_2])$ would be expected to vary between
around 0.3 – 1, assuming a daytime $[HO_2] = 2 \times 10^7$ molecules $cm^{-3}$ (winter time – the alkyl
nitrate and alkane signals in the firn are dominated by winter time concentrations). Changes to
[NO] or $[HO_2]$ in these remote environments will affect the production efficiency of the alkyl
nitrates. Since the term $\gamma$ is an average across the whole transport time it reflects both the urban
and remote environments.
Equation E6 is a rearrangement of Equation E5 from which historic changes to $\gamma$ can be
calculated using the measured changes to the $[RONO_2]/[RH]$ ratio (assuming that the
photochemical processing, $[\overline{OH}]t$, has remained constant through time). All rate constants and
branching ratios used in the calculations are taken from MCMv3.3.1 (mcm.leeds.ac.uk; Jenkin
et al., 1997) (see Table S2, Supplementary Information) assuming a temperature of 273 K. The
magnitude of the diurnal photolysis sink of the alkyl nitrates, $j_{16}$, will vary with emission region
and during transport. $j_{16}$ is included in the term $\lambda$, which represents the ratio $j_{16}/k_{15}[\overline{OH}]$. $\lambda$ is
assumed to be 1 in Figure 4, i.e. $j_{16} = k_{15}[\overline{OH}]$. The sensitivity to the magnitude of this sink is
discussed further in Supplementary Information.

$$\gamma = \frac{[RONO_2](k_{15}(1+\lambda)-k_{13})}{[RH]\beta k_{13}(1-e^{(k_A-k_B)t})} \qquad \text{(E6)}$$

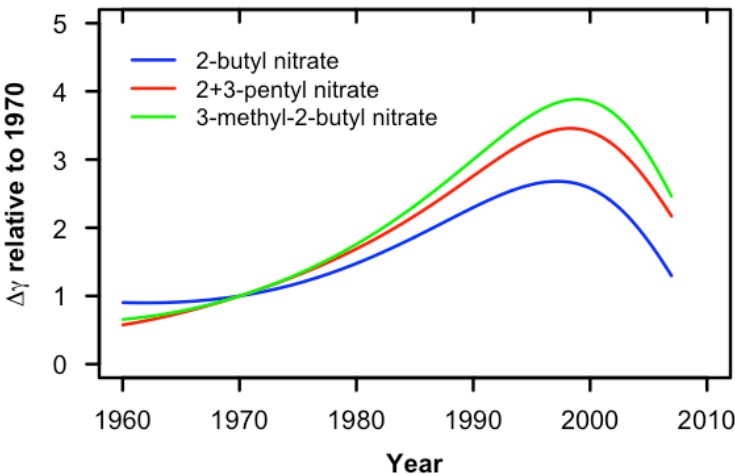

Figure 4 The trend in the mean alkyl nitrate production efficiency, γ, of the air masses in which the alkyl
nitrates were formed, calculated using Equation E6 for each of three alkyl nitrate/alkane pairs, relative to
1970 values. This assumes that the amount of photochemical processing, $\overline{[OH]}t$, remained constant at $5\times10^{11}$
molecules cm$^{-3}$ s. All rate constants and branching ratios used in the calculations are taken from MCMv3.3.1
(mcm.leeds.ac.uk) assuming a temperature of 273 K. The mean diurnally averaged photolysis sink $j_{16}$ was
assumed to be equal to the mean OH sink, $k_{15}\overline{[OH]}$.
Figure 4 shows the historical trend in mean alkyl nitrate production efficiency, γ, relative to
1970, calculated using Equation E6 if $\overline{[OH]}t$ is assumed to have remained constant during this
period. A value of $5\times10^{11}$ molecules cm$^{-3}$ s is used for the constant $\overline{[OH]}t$. This is based on a
mean transport time of air masses from Europe (from where the majority of winter-time
pollutants are transported to the Arctic – see Section 6.1) to the Arctic in the winter of ten days
(Stohl, 2006), and a mean winter-time [OH] of ~ $6\times10^{5}$ cm$^{-3}$ (in reasonable agreement with that
derived by Derwent et al. (2012) for the North Atlantic in winter-time). However, it is noted
that the relative change in γ shown in Figure 4 is independent of the value used for $\overline{[OH]}t$.
The trend in the mean production efficiency of the alkyl nitrates, γ, relative to 1970 values,
shows similar features to those of the alkyl nitrate trends. The ratio increases by a factor of
between 2.5 (2-butyl nitrate) and 4 (3-methyl-2-butyl nitrate) from the 1970s to the late 1990s
and then subsequently declines.
This analysis suggests that the observed changes to the [RONO$_2$]/[RH] ratio in the firn could
be explained by changes to γ. This is driven by changes to the [NO]/[HO$_2$] ratio experienced by
air masses in transport to the Arctic. We now investigate whether trends in processes that could
drive this ratio are consistent with this scenario, i.e. how $NO_X$ concentrations may have
changed.

## 5.1 Changes to atmospheric $NO_X$ concentrations

### 5.1.1 $NO_X$ sources

The atmospheric $NO_X$ concentration is determined by the relative magnitudes of the sources
and sinks. The main sources of $NO_X$ in the northern hemisphere are anthropogenic emissions
from fossil fuel use, power stations and transport (Olivier and Berdowski et al., 2001; Olivier
et al., 2001).
Figure 5 shows how $NO_X$ emissions from North America, Europe and Russia have varied
between 1970 and 2008, taken from the bottom-up estimates of the EDGAR database (EDGAR
v4.2, http://edgar.jrc.ec.europa.eu). Emissions were fairly constant between 1970 and 1990 and
then fell by about 25 % from 1990 to 2008. This is in good agreement with the $NO_X$ emission
trends for OECD Europe presented in Vestreng et al. (2009).

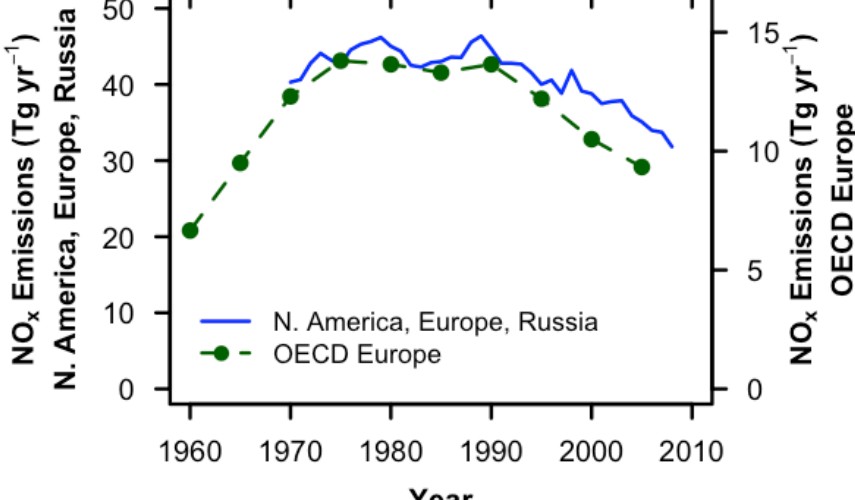

Figure 5 Blue solid line (left axis): The trend in $NO_X$ emissions (Tg yr$^{-1}$) from North America, Europe and Russia
for the period 1970 to 2008 (EDGAR v4.2, http://edgar.jrc.ec.europa.eu). Green points (and dashed line) (right
axis): OECD Europe $NO_X$ emissions (Tg yr$^{-1}$) from Vestreng et al. (2009).

Assuming that these bottom up emissions estimates are correct in the timing of the $NO_X$ emissions changes, it seems unlikely that an increase in the alkyl nitrate production efficiency during the period 1970 to the late 1990s could have been driven primarily by changing $NO_X$ emissions.

A decline in the alkyl nitrate production efficiency after the late 1990s, on the other hand, may well have been driven by decreasing $NO_X$ emissions. Measurements at a range of UK sites showed a decrease in $NO_X$ concentrations from 1996 (the beginning of the reported measurements) to 2002 – 2004, of 1 – 3.5 % per year, depending on the site (Carslaw et al., 2011). During the same period, the fraction of the $NO_X$ that is $NO_2$ (f-$NO_2$) roughly doubled – suggesting that NO has decreased by more than $NO_x$ concentrations. Recent trends at many European sites show similar trends with small decreases in $NO_x$ between 1999 and 2007 (the period for which measurements are available) but level or increasing $NO_2$ through the same period (Carslaw et al., 2011; Gilge et al., 2010).

Declining $NO_X$ emissions have been used to explain these trends in measured concentrations. However, the decline in these ambient concentrations is not as large as would be expected using current emission inventories (Carslaw et al., 2011).

## 5.1.2 $NO_X$ sinks

At mid-high latitudes, in the daytime, during the summer, the main sink for $NO_X$ is the reaction of $NO_2$ with OH. This reaction produces nitric acid ($HNO_3$), much of which is then removed from the atmosphere by wet deposition. However, at night and during the winter months, when daily mean [OH] is more than an order of magnitude lower than during the summer (e.g. Derwent et al., 2012), the dominant $NO_X$ sink is conversion of dinitrogen pentoxide ($N_2O_5$) to $HNO_3$. This only occurs when photolysis is low, allowing $NO_3$ (formed from the reaction of $NO_2$ with $O_3$) to build up. This $NO_3$ reacts with $NO_2$ to form $N_2O_5$. While the reaction of $N_2O_5$ with $H_2O$ is slow in the gas-phase (Tuazon et al., 1983), it occurs rapidly in aerosol.

A modelling study by Dentener and Crutzen (1993) predicted that changes to the loss of $NO_x$ via sulfate aerosol could have a significant effect on northern hemisphere $NO_x$ concentrations and that these changes would also affect $O_3$ and OH concentrations. Subsequent modelling studies, though often focussing on remaining uncertainties in the uptake coefficients of $N_2O_5$ to aerosol, have broadly agreed with the magnitude of the $NO_X$ changes suggested by Dentener and Crutzen (Brown and Stutz, 2012).

There has been a large decrease in sulfate aerosol observed in Europe and the United States
since 1980 (Berglen et al., 2007; Turnock et al., 2015). Figure 6 shows the measured trend in
winter-time (DJF) sulfate mass concentration presented in Turnock et al. (2015), with decreases
of about 75% from 1979 to 2005. This decreasing trend has been driven by a ~70 % decrease
in $SO_2$ emissions (Smith et al., 2011) from these regions (Figure 6). It is noted that while global
$SO_2$ emissions have only decreased about 15 % from the peak in the 1970s, due to rapidly
increasing emissions in East Asia in recent decades, sulfate aerosol has a lifetime of about 5
days in the troposphere (and $SO_2$ of about 1 day) (Stevenson et al., 2003) and so aerosol
concentrations will be largely driven by regional $SO_2$ emissions.

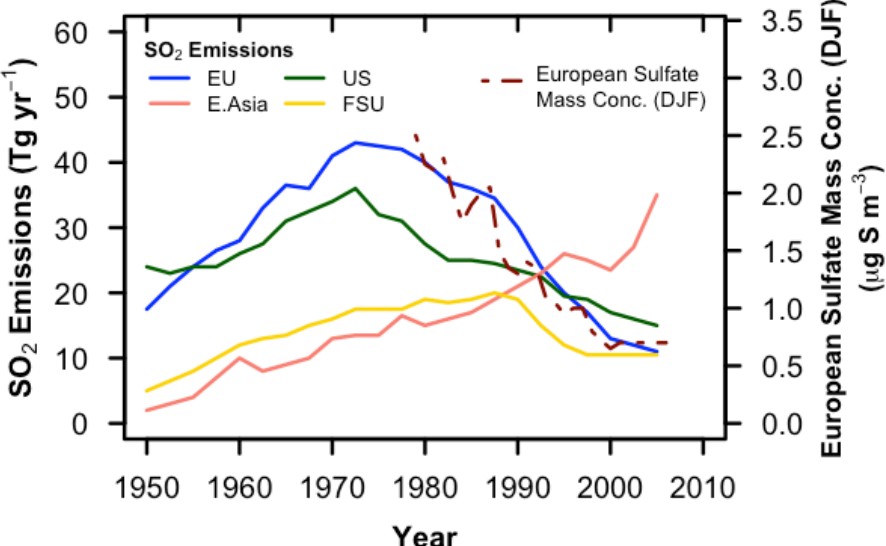

Figure 6 $SO_2$ emissions (Tg yr$^{-1}$) 1950-2005 from Smith et al. (2011), and mean European sulfate mass
concentration ($\mu$g S m$^{-3}$) in winter (DJF) from Turnock et al. (2015). $SO_2$ emissions: Blue – Europe; Green – N.
America (US + Canada); Gold – Former Soviet Union (Russia, Ukraine, others); Pink – E. Asia (China, Japan, S.
Korea, others). Brown dashed line - mean European sulfate mass concentration in winter (DJF).
These large decreases in sulfate aerosol in Europe and the US (the main source regions for air
masses arriving in the Arctic in the winter) may be expected to have led to a decrease in $NO_X$
removal by $N_2O_5$ hydrolysis, and hence to an increase in the $NO_X$ lifetime and atmospheric
[$NO_X$] either through long term changes to the total sulfate aerosol burden (Turnock et al., 2015)
or long term changes to particle acidity driven by reductions in sulfate (e.g. Murphy et al.,
2017). The time period of decreasing $SO_2$ emissions and sulfate aerosol is broadly in line with
the derived steep increase in the alkyl nitrate production efficiency.
However, work remains ongoing to determine the exact effect of a number of parameters (e.g.
relative humidity, particulate organic / sulfate ratio, particle acidity) on the $N_2O_5$ uptake
coefficient and thus the efficacy of the reaction and the extent to which changing sulfate content
and abundance of aerosol would be expected to affect the uptake coefficient and thus oxidant
concentrations (Brown and Stutz, 2012).

## 6   Changes to Photochemical Oxidation

An alternative explanation for the observed alkyl nitrate trends is that the amount of
photochemical processing of the air masses reaching the Arctic changed during the period of
study. An increase in processing could be caused by a change in either the concentration of the
OH radical (assuming photolysis to have remained constant), or by an increase in the transport
time of the air mass from the source region to the Arctic.
Equation E7 is a rearrangement of Equation E6 from which historic changes to the
photochemical processing, $\overline{[OH]}t$, can be calculated using the measured changes to the
$[RONO_2]/[RH]$ ratio, assuming that $\gamma$ has remained constant.
$$\overline{[OH]}t = ln\left(1 - \frac{[RONO_2](k_{15}(1+\lambda)-k_{13})}{[RH]\gamma\beta k_{13}}\right) \div \left(k_{13} - k_{15}(1+\lambda)\right) \qquad (E7)$$

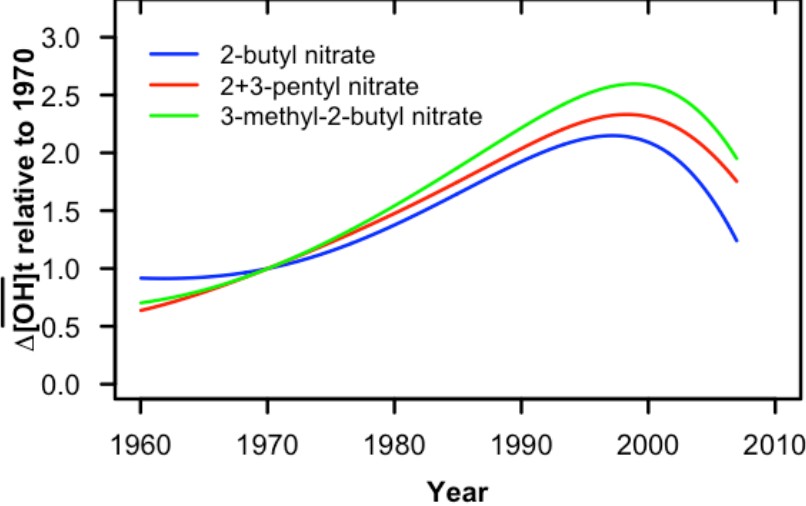

Figure 7 The trend in $\overline{[OH]}t$ calculated using Equation E7 for each of three alkyl nitrate-alkane pairs assuming a
constant value for $\gamma$.
Figure 7 shows the trends in $\overline{[OH]}t$ derived from the alkyl nitrate-alkane pairs if a constant value
for γ is assumed. The value used for the constant γ for each alkyl nitrate was the mean value
derived in Figure 4 for the period 1960 – 2007 (0.31 for 2-butyl nitrate, 0.34 for 2+3-pentyl
nitrate, 0.17 for 3-methyl-2-butyl nitrate).
Equation E7 also has [OH] terms on the right hand side of the equation, incorporated in λ. The
results in Figure 7 are determined through an iterative process of fitting a polynomial to the
trend calculated using an *a-priori* assumption that λ = 1 for the whole time period. The
calculation is then repeated with a temporally varying value for λ using this fit to determine the
changes. This process converges towards the unique solution presented in Figure 7. E.g. In
Figure 7 for 2-butyl nitrate, in 1970 the assumed value of λ is 1, at the peak of $\Delta\overline{[OH]}t$ in 1997,
when $\Delta\overline{[OH]}t = 2.1$, the value of λ is 0.46 (1/2.1).
Figure 7 shows that the observed [RONO$_2$]/[RH] ratios between around 1970 and the late 1990s
could be explained by a relative change in $\overline{[OH]}t$ of a factor of between 2.1 (2-butyl nitrate) and
2.6 (3-methyl-2-butyl nitrate). The sensitivity of these calculated values to the assumed value
for γ and for $j_{16}$ in 1970 are discussed in the Supplementary Information.

## 6.1 Air mass transport time to the Arctic

The transport time, *t*, of pollutants to the Arctic from source is dependent on (i) the atmospheric
transport patterns, and (ii) the source regions of the pollutants.
Concerning (i), Kahl et al. (1999) have suggested that there is a decadal scale (4 – 14 years)
variability in transport patterns of pollutants from the NH to the Arctic but note no long term
trend. Hirdman et al. (2010) note that while changes to transport patterns can explain much of
the inter-annual variability of Arctic concentrations of black carbon and sulfate aerosol
(pollutants with similar source regions to the alkanes), they played only a minor role in long
term changes. Eckhardt et al. (2003) have shown that transport of pollutants to the Arctic from
European and US source regions is more rapid during positive phases of the weather pattern,
the North Atlantic Oscillation (NAO). During the period 1960 – 1980 the NAO was
predominantly in a negative phase in winter, between 1980 and 2000 it was predominantly in a
positive phase, and since 2000 neither phase has been dominant (Hurrell and Deser, 2010). This
suggests that there was more rapid transport of pollutants to the Arctic during the period 1980
– 2000 compared to the preceding and succeeding periods. This would mean a shorter

processing time for the air masses and hence less alkyl nitrate production and lower alkyl nitrate to alkane ratios. This is the opposite to what we observe in the firn records, suggesting that changes to the NAO are unlikely to be responsible for the observed alkyl nitrate trends.

Concerning (ii), changes to the relative distribution of the major source regions of the alkanes could have occurred for a number of reasons. Fuel composition has changed through time as a response to technological development of vehicles. Clean air legislation has led to the development of cars with progressively lower evaporative and tailpipe emissions (e.g. Wallington et al., 2006), through developments such as catalytic converters. In addition emissions may have changed simply due to a change in vehicle usage. If such changes were to have occurred in more northerly regions significantly earlier than in more southerly regions, this could have increased the mean transport time of air masses to the Arctic.

For many areas in North America, the Reid vapour pressure of fuel is regulated in the summer season (June 1 - September 15) (epa.gov.uk), leading to sale of a different fuel mix in summer compared to winter. This is generally achieved by producers reducing the butane content of the fuel (Gentner et al., 2006). This legislation came in in 1990. However, the observed alkane and alkyl nitrate signals in Greenland are almost entirely winter-time signals (e.g. Swanson et al., 2003), and so such seasonal variation in fuel composition would not be expected to affect the firn measurements.

The main sources of anthropogenic emissions to the Arctic of gases with lifetimes on the order of a few weeks, particularly during the winter, have been identified as being northern Eurasia (*e.g.* Shindell et al., 2008; Stohl et al., 2007; Klonecki et al., 2003). Emissions from Europe and North America have followed a similar declining trend in recent years (Lamarque et al., 2010; von Schneidemesser et al., 2010; Warneke et al., 2012;), thus the relative contribution from each region is not expected to have changed dramatically.

## 6.2   The Hydroxyl Radical, OH

The alternative explanation for an increase in photochemical processing is an increase in the mean [OH] to which the air mass is exposed. This would represent a regional trend in [OH] representative of regions from and through which air masses are transported to the Arctic, and would relate primarily to the winter (since the alkyl nitrate and alkane signals in the firn are dominated by winter time concentrations). It is noted that the increased chemical processing

observed could also result from an increase in an oxidant other than OH, e.g. atomic chlorine,
as suggested in Helmig et al. (2014b).
Studies using changes to atmospheric mixing ratios of methyl chloroform ($CH_3CCl_3$) have
suggested that global mean OH concentrations are 'well buffered' (e.g. Montzka et al., 2011).
Since the main sink of $CH_3CCl_3$ in the atmosphere is reaction with OH, and the emission
sources and other sinks are thought to be well constrained, the variation in its observed mixing
ratios at a number of remote sites can be used to infer variations in global [OH]. Global mean
[OH] has been inferred in this way in a number of studies (Prinn et al., 1995, 2001, 2005; Rigby
et al., 2008; Montzka et al., 2011). The most recent of these (Montzka et al., 2011) reported
little inter-annual variability in mean global atmospheric [OH] estimating roughly 5% variation
from the mean value during the period 1997 – 2008, but this does not cover the period of interest
here (1970 to later 1990s). Similarly, Kai et al. (2011) inferred a low variability in global [OH]
between 1998 and 2005 based on a relatively constant δ-D-$CH_4$. Earlier work using methyl
chloroform (Prinn et al., 2001) reported an increase in NH [OH] of roughly 40% between 1979
and 1991 but this increase has been questioned in more recent work (*e.g.* Montzka et al., 2011).
However, there are a growing number of observational data sets of trace gases in the NH which
show trends since 1980 that could be explained, at least in part, by changes to the concentration
of the OH sink. E.g. decreasing Arctic alkane mixing ratios (Helmig et al., 2014b; Aydin et al.,
2011); decreasing Arctic CO mixing ratios (Petrenko et al., 2013); increasing $d^{13}C$ of methane
(Monteil et al., 2011; Sapart et al., 2013); decreasing $dC^{16}O$ of Arctic CO (Wang et al., 2012).
A recent multi-model inter-comparison exercise of seventeen global chemical transport models,
showed agreement for a small increasing trend in global mean [OH] of 3.5 (± 2.2) % between
1980 and 2000 and a slightly larger [OH] increase in the northern hemisphere of 4.6 (± 1.9) %
(Naik et al., 2013). Dalsøren et al. (2015) determined an increase in global mean [OH] of about
10 % between 1970 and 2006 from modelled increases of the methane lifetime.

## 7 Discussion

The alkyl nitrate trends presented herein suggest a profound change to the chemical state of the
northern hemisphere mid-high latitude atmosphere in winter between the 1970s and the late
1990s and then again between the late 1990s and the mid-2000s.
A key species of the tropospheric chemistry cycle, tightly linked to $NO_X$ and $HO_X$, is ozone
(Figure 1). Ozone mixing ratios increased at background sites across the NH during the second
half of the twentieth century, roughly doubling since 1960 (Parrish et al., 2012). Ozone
production is positively linked to $[NO_X]$ (at low $NO_X$ concentrations such as the background
atmosphere). Hence, an increase in the $[NO]/[HO_2]$ ratio from around 1970 to the mid-1990s is
consistent with long-term trends in ozone in the background atmosphere.
Furthermore, while the alkyl nitrate measurements represent changes to the winter-time
atmosphere, the ozone trends are seen in both summer and winter. If these are being driven by
increases to $[NO_X]$ in the background atmosphere, then this suggests that the chemical changes
to the atmosphere may have been present throughout the year and are not just a winter time
phenomenon.
This work also implies that there may have been a change in [OH]. Indeed due to the connected
nature of the chemistry of $NO_X$, ozone and OH (Figure 1) it seems unlikely, given the implied
increases in $NO_X$ suggested here, and the recorded increases in ozone (Parrish et al., 2012), that
there was not a commensurate increase in OH production during this period. A major primary
production route of OH is via photolysis of ozone (Equation E8 - Smith et al., 2006).

17       $P(OH) = 2f[O_3] \times j(O^1D)$                                          (E8)

Where P(OH) is primary production of OH, and $f$ is the fraction of $O(^1D)$ that reacts with water
vapour. Ozone has increased at background sites between 1960 and 2000 (Parrish et al., 2012)
and measured water vapour has also increased slightly (Hartmann et al., 2013). It therefore
seems that the primary production of OH in the background atmosphere from this source must
have increased through the final decades of the past century. Another primary OH source is via
ozonolysis of alkenes (Johnson and Marston, 2008). A third source of OH that may be important
is photolysis of HONO (e.g. Stone et al., 2012). There is still considerable uncertainty about
the sources of HONO, with formation from heterogeneous conversion of $NO_2$ via a range of
postulated processes appearing to dominate over the $HO_X$ / $NO_X$ recycling reaction OH + NO
(e.g. Michoud et al., 2014). This again would be a primary source of OH which would be
expected to correlate positively with $NO_X$ concentrations.
The primary sink of OH in the background atmosphere, CO, has decreased by about 15% since
1980 (Petrenko et al., 2013), with the secondary sink, $CH_4$, having increased between 1980 and
2000 by about 15%.

The global growth rate of methane in the atmosphere continually declined throughout the period of the 1970s to 2000, culminating in the 'methane pause' between 1999 and 2006 (Dlugockencky et al. 2009). A possible cause of this change in growth rate in methane could be an increase in OH concentration (e.g. Dalsøren et al. 2016). This is consistent with an increase in [OH] also being the cause of the trend in the ratio of alkyl nitrate to parent alkane seen in this work. It should be noted, though, that the majority of the OH oxidation of methane occurs in the tropics (e.g. Bloss et al. 2005), while any increase in OH suggested by the work herein must be viewed as representative only of the mid-high latitude northern hemisphere and the winter time. However, the processes suggested herein, such as changes to the $N_2O_5$ sink, have been shown to be effective at a hemispheric scale.

In a recent inter-model comparison project, ACCMIP, it was shown that models failed to capture the measured magnitude of the increase in ozone over recent decades (Parrish et al., 2014), in particular the steep increase seen between 1980 and 2000. This failure to capture measured changes to ozone may suggest that models are likely to under-estimate changes to OH production, from ozone photolysis or reactions of ozone with alkenes, over the same period (i.e. they may be larger than the 4.6 (± 1.9) % reported in Naik et al. (2013) for the NH).

Including alkyl nitrate chemistry and using the alkyl nitrate measurements presented herein could provide a valuable constraint for global chemical transport models modelling changes to NOx and HOx back to the middle of the twentieth century.

## 8   Conclusions

Time series such as those presented here are fundamental to improving our understanding of trends in atmospheric composition during the twentieth century. The long-term trends of alkyl nitrates presented herein suggest major changes to the chemical state of the atmosphere during the past five decades. The observed large increase in the $[RONO_2]/[RH]$ ratio between the 1970s and late 1990s could be explained by a 2 - 4 fold increase in the mean production efficiency of the alkyl nitrates, driven by an increase in the $[NO]/[HO_2]$ ratio in the background atmosphere. This is not consistent with reported changes to northern hemisphere $NO_X$ emissions, but may have been driven by a reduction in the $NO_X$ sink. The recent decreases (since the late 1990s) in alkyl nitrate concentrations are in qualitative agreement with recent decreases in $NO_X$ emissions and in measured $NO_X$ concentrations. Alternatively, the observed increase in the $[RONO_2]/[RH]$ ratio between the 1970s and late 1990s could be explained by

an increase in the amount of photochemical processing [OH]$t$ of air masses reaching the Arctic by a factor of 2 – 3. This could be driven by an increase in concentrations of the hydroxyl radical (OH), or to the transport time ($t$) of air masses from source regions to the Arctic. If the observed trends are driven by changes to the chemical state of the atmosphere, then it is likely that they represent a combination of changes to the concentrations of both NO and OH.

**Acknowledgements**

This work was supported by funding from the UK Natural Environment Research Council (NE/F021194/1 & NE/M003248/1). NEEM is directed and organized by the Centre of Ice and Climate at the Niels Bohr Institute and US NSF, Office of Polar Programs. It is supported by funding agencies and institutions in Belgium (FNRS-CFB and FWO), Canada (NRCan/GSC), China (CAS), Denmark (FIST), France (IPEV, CNRS/INSU, CEA and ANR), Germany (AWI), Iceland (RannIs), Japan (NIPR), South Korea (KOPRI), The Netherlands (NWO/ALW), Sweden (VR), Switzerland (SNF), the United Kingdom (NERC) and the USA (USNSF, Office of Polar Programs) and the EU Seventh Framework programs. We are indebted to Jakob Schwander of the Physics Institute at the University of Bern, Switzerland for collecting the firn air samples at NEEM, and Thomas Blunier of the Centre for Ice and Climate at University of Copenhagen, Denmark for leading the NEEM gas consortium. We thank Chelsea Thompson for useful discussions.

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
