# Peer review of "Changes to the chemical state of the northern hemisphere"

_Atmospheric Chemistry and Physics, 2016_

## Referee Comment (RC1) · Anonymous Referee #2 · 4 Jan 2017

General comments. This manuscript is an ambitious work that derives alkyl nitrate trends from firn air measurements and combines them with similarly-derived (already published) parent alkane data to characterize possible scenarios for large changes in the chemical state (especially the [NO]/[HO2] ratio) of the northern hemisphere during the 1970s to late 1990s. The manuscript explores the possibility that the data could indicate significant changes in NOx sources and sinks (driven by trends in sulfate aerosol precursors). Other potential explanations (such as changes in OH and transport) are also considered. The manuscript makes some very wide-ranging/large scale conjectures using this relatively limited new data. However, I consider that the authors are careful to back up their suggestions where possible and provide appropriate caveats

where due. I recommend that the manuscript be accepted basically "as is", but I request that the authors consider making minor revisions as suggested below.

Specific comments. Given the potential importance of the trends in SO2 emissions and their implied influence on the NOx budget (as outlined in section 5.1.2) I suggest that this topic be represented in a little more depth.

Specific suggestions. Section 5.1.2 NOx sinks: Given the potential significance of this section to the overall conclusions, it seems to be somewhat truncated compared to other parts. As the timing of the various trends appear to be crucial to the various evidence lines presented in this work, maybe the authors could add a figure (or perhaps combine with existing Fig 5?) to convey the time series for sulfate aerosol trends (eg like in Fig 8 in Smith et al., 2011) to more concretely illustrate when sulfur emissions peaked.

Technical corrections typing errors, etc. Page 19, line 17 – I do not find the "Aydin et al." reference in the reference section Page 23, Line 23 – I do not find the "Dlugokencky et al." reference in the text.

---

## Referee Comment (RC2) · Anonymous Referee #1 · 13 Jan 2017

Summary:

This manuscript presents a very valuable data set of historical changes in alkyl nitrates and their parent hydrocarbons. Assuming 1) that the measurements of the concentrations of these species in the firn air are valid, and 2) that the conversion of firn air concentrations as a function of depth to ambient concentrations as a function of year are valid, then the results of Sections 1 through 3 are worthy of publication. However, the analysis of the ratios of alkyl nitrate to parent alkane is flawed and does not justify the very far reaching conclusions given in this paper, i.e., the analysis does not establish that the chemical state of the northern hemisphere atmosphere changed during the second half of the twentieth century. The analysis, discussion and conclusions in

[Figure]

Sections 4-8 must be reconsidered and modified as necessary. When the analysis is reconsidered, it may not be possible to reach significant general conclusions. Specific issues are detailed below.

Major issues:

1) The fatal problem with this paper is that the derivation of Equations E2 and E3 is flawed. Thus, the conclusions of this paper that are based on these equations are also flawed. For this paper to be acceptable for publication, the authors must reconsider all analyses based on these equations, and revise the conclusions accordingly. Bertman et al. [1995] derived Equation E1 by integrating the differential equation for the time rate of change of the alkyl nitrate concentration, which required approximations that were valid in an NOx rich environment. To extend Equation E1 to yield Equation E2 as the authors have done is not valid. To be valid, the original differential equation of Bertman et al. [1995] must be properly modified to include Reaction 11, and then properly integrated.

As it stands, it is clear that Equation E2 cannot be correct. The left side of the equation is a ratio that is determined by long time scale evolution, while the right side depends on NO and HO2 concentrations, which are variables that vary on very short time scales. For example, if the NO concentration were suddenly increased by fresh emissions, Equation E2 indicates that the [RONO2]/[RH] ratio would change, which is obviously incorrect. It might be (or might not be) that a proper integration of the appropriate differential equation could yield an equation that looks like Equation E2, where the NO and HO2 concentrations would be weighted time integrals, but those weightings are likely to be different for each species. How those weightings might be calculated is not clear. Equation E3 is derived from Equation E2, so the problems in the latter propagate to the former. The left side is now a ratio of highly variable species, while the right side depends on concentrations that change relatively slowly. The authors are evidently implicitly assuming that the NO and HO2 concentrations are some sort of remote, hemisphere wide average (as indicated by the statement on lines 11-12 of page 13).

However, this is far from clearly correct, as the production of the alkyl nitrates may well be dominated by relatively rapid photochemistry in the urban, NOx-rich, higher OH environment, rather than photochemistry in the remote environment. Hence, properly integrating the differential equations describing the photochemical processes is critical to correct analysis. That integration may not be possible to carry out analytically.

2) It is unclear whether the uncertainties in the estimated diffusion coefficients of the alkyl nitrates have an important impact on the atmospheric history reconstructions. This should be discussed.

3) On page 7 "It is noted that the latter part (post-1995) of the model derived scenarios for 2+3-pentyl nitrate is rather sensitive to the inclusion or exclusion of the measurement at 34.72 m (the most shallow measurement used). The scenarios presented in this work are based on including this measurement." Is this an important issue? The authors should discuss the effect on the final conclusions if this measurement is excluded.

Minor issues:

1) Abstract lines 18-21 - I disagree with the statement "Due to their short atmospheric lifetimes, NOx and HOx are highly variable in space and time, and so the measurements of these species are of very limited value for examining long term, large scale changes to their budgets." If measurements of either HOx or NOx were measured in a given region in an extensive enough manner to characterize average ambient concentrations (e.g., during an intensive aircraft deployment such as NASA DC-8 studies), and if those extensive measurements were repeated after the passage of a decade or so, then it should be possible to quantify the long term, large scale changes to their concentrations. I suggest that this statement be removed, as it is superfluous to the paper.

2) Page 2, lines 13 and 14. I also disagree with the phrase "which is positively correlated with NOx concentrations in the background atmosphere through the photolysis

of NO2 (Reactions R1-R2)." Ozone is certainly correlated with total NOy concentrations, but since ozone and NOx have very different lifetimes, they are in general only poorly correlated, because any correlation resulting from production through the photolysis of NO2 is destroyed by removal or fresh emissions of NOx, even in the remote troposphere. Here and elsewhere throughout the paper, I suggest that very careful consideration be given to the veracity of each sentence.

3) Page 2, lines 23 and 24 - The phrase "The main removal processes for HOx are the reaction of OH with NO2 (Reaction R9) ..." has an error. Evidently this reaction has not been separately listed in the paper.

4) Page 4, line - Should be "on-road" vehicles.

5) Page 6 - Sentence on lines 18-21 is not clear.

6) Page 8 - Line 12 - The Worton et al. (2010) reference is not in the References list; should be Worton et al. (2012)?

7) Page 12, lines 19-22 are not exactly correct: In Equation E1, taken from Bertman et al. (1995), [OH] is assumed to be the average over the time t. This is exactly what the authors are doing as well. This should be clarified, i.e. the term [OH]t represents the time integral of [OH] over the period during transport from the source region to the Arctic.

---

## Author Comment (AC1) · 28 Feb 2017

**Response to Reviewer #2 of submission of:**

**Changes to the chemical state of the northern hemisphere atmosphere during the second half of the twentieth century by Newland et al., 2017, submitted to ACPD**

**General Response**

We thank the referee for giving their time to make comments helping to clarify and improve our manuscript. Responses to each point are given separately beneath that point. The referee's comments are bold and italic, the author's comments are inset in plain type.

*Specific comments. Given the potential importance of the trends in SO2 emissions and their implied influence on the NOx budget (as outlined in section 5.1.2) I suggest that this topic be represented in a little more depth.*

*Specific suggestions. Section 5.1.2 NOx sinks: Given the potential significance of this section to the overall conclusions, it seems to be somewhat truncated compared to other parts. As the timing of the various trends appear to be crucial to the various evidence lines presented in this work, maybe the authors could add a figure (or perhaps combine with existing Fig 5?) to convey the time series for sulfate aerosol trends (eg like in Fig 8 in Smith et al., 2011) to more concretely illustrate when sulfur emissions peaked.*

     Author response:

     We have included a new Figure in the manuscript (Figure 6) to show the $SO_2$ emissions 1950 – 2005 presented in Smith et al. (2011, ACP) and observed winter-time sulfate aerosol mass concentration in Europe (Turnock et al., 2015, ACP).

     We have altered the text of Section 5.1.2 to reflect the inclusion of this figure and have been a little more explicit in the regional nature of sulfate aerosol in relation to $SO_2$ emissions.

[Figure]

**Figure 6** $SO_2$ emissions (Tg yr$^{-1}$) 1950-2005 from Smith et al. (2011), and mean European sulfate mass concentration ($\mu$g S m$^{-3}$) in winter (DJF) from Turnock et al. (2015). $SO_2$ emissions: Blue – Europe; Green – N. America (US + Canada); Gold – Former Soviet Union (Russia, Ukraine, others); Pink – E. Asia (China, Japan, S. Korea, others). Brown dashed line - mean European sulfate mass concentration in winter (DJF).

"A modelling study by Dentener and Crutzen (1993) predicted that changes to the loss of $NO_x$ via sulfate aerosol could have a significant effect on northern hemisphere $NO_x$ concentrations and that these changes would also affect $O_3$ and OH concentrations. Subsequent modelling studies, though often focusing on remaining uncertainties in the uptake coefficients, have broadly agreed with the magnitude of the changes suggested by Dentener and Crutzen (Brown and Stutz, 2012).

There has been a large decrease in sulfate aerosol observed in Europe and the United States since 1980 (Berglen et al., 2007; Turnock et al., 2015). Figure 6 shows the measured trend in winter-time (DJF) sulfate mass concentration presented in Turnock et al. (2015), with decreases of about 75% from 1979 to 2005. This decreasing trend has been driven by a ~70 % decrease in $SO_2$ emissions (Smith et al., 2011) from these regions (Figure 6). It is noted that while global $SO_2$ emissions have only decreased about 15% from the peak in the 1970s, due to rapidly increasing emissions in East Asia in recent decades, sulfate aerosol has a lifetime of about 5 days in the troposphere (and $SO_2$ of about 1 day) (Stevenson et al., 2003) and so aerosol concentrations will be largely driven by regional $SO_2$ emissions.

These large decreases in sulfate aerosol in Europe and the US (the main source regions for air masses arriving in the Arctic in the winter) would be expected to have led to a decrease in $NO_x$ removal by $N_2O_5$ hydrolysis, and hence to an increase in the $NO_x$ lifetime and atmospheric [$NO_x$]. The time period of decreasing $SO_2$ emissions and sulfate aerosol is broadly in line with the derived steep increase in the [NO]/[HO$_2$] ratio."

Stevenson, D. S., C. E. Johnson, W. J. Collins, and R. G. Derwent, The tropospheric sulphur cycle and the role of volcanic SO2, in Volcanic Degassing edited by C. Oppenheimer, D. M. Pyle and J. Barclay, Geol. Soc. Lond. Spec. Pub., 213, 295-305, 2003.

***Technical corrections typing errors, etc.***

***Page 19, line 17 – I do not find the "Aydin et al." reference in the reference section***

Author response:

Reference added.

***Page 23, Line 23 – I do not find the "Dlugokencky et al." reference in the text.***

Author response:

I think this was just a case of mis-spelling. Dlugokencky is now spelled correctly throughout the manuscript and appears in the text and references.

**Additional Changes in Response to both Referees' Reviews**

The following sentence has been added to the end of paragraph 1 of Section 5:

*"Since the term $k_{14}[NO]/(k_{14}[NO]+k_{11}[HO_2])$ is an average across the whole transport time it reflects both the urban and remote environments."*

The last paragraph of Section 5.0 has changed from:

*"To investigate the drivers that might have led to these changes in [NO]/[HO2] ratio, we shall now examine how the NOx and $HO_2$ concentrations may have changed."*

to:

*"This analysis suggests that the observed changes to the $[RONO_2]/[RH]$ ratio in the firn could be explained by changes to the average $[NO]/[HO_2]$ ratio experienced by air masses in transport to the Arctic. We now investigate whether trends in processes that could drive this ratio are consistent with this scenario, i.e. how $NO_x$ and $HO_2$ concentrations may have changed."*

---

## Author Comment (AC2) · 28 Feb 2017

**Response to Reviewer #1 of submission of:**

**Changes to the chemical state of the northern hemisphere atmosphere during the second half of the twentieth century by Newland et al., 2017, submitted to ACPD**

**General Response**

We thank the referee for giving their time to make comments helping to clarify and improve our manuscript. Responses to each point are given separately beneath that point. The referee's comments are bold and italic, the author's comments are inset in plain type.

With respect to the primary concern of the referee, we are not sure whether there has been some misunderstanding over the nature of our use of the ratio $k_{14}[NO]/(k_{14}[NO]+k_{11}[HO_2])$ or whether it is that the referee has concerns over the validity of the use of an average value for this ratio. We hopefully address both concerns in our responses.

While we present a lot of work in response to the referee's major concern, we do not feel that much of it belongs in the paper beyond clarifying that the ratio $k_{14}[NO]/(k_{14}[NO]+k_{11}[HO_2])$ is representative of a mean value during transport.

*Major issues:*

*1) The fatal problem with this paper is that the derivation of Equations E2 and E3 is flawed. Thus, the conclusions of this paper that are based on these equations are also flawed. For this paper to be acceptable for publication, the authors must reconsider all analyses based on these equations, and revise the conclusions accordingly. Bertman et al. [1995] derived Equation E1 by integrating the differential equation for the time rate of change of the alkyl nitrate concentration, which required approximations that were valid in an NOx rich environment. To extend Equation E1 to yield Equation E2 as the authors have done is not valid. To be valid, the original differential equation of Bertman et al. [1995] must be properly modified to include Reaction 11, and then properly integrated.*

> The ratio $k_{14}[NO]/(k_{14}[NO]+k_{11}[HO_2])$ as used in Equation E2 (and subsequent rearrangements) represents an average over time, t, in the same way as [OH], which is embedded in the $k_A$ and $k_B$ terms in the Bertman et al equations. When referring to this ratio as an average over time in the following discussion we shall label it $\gamma$.
>
> Below we show how we get from the rate equation for [RONO$_2$] presented in Bertman et al., to Equation E2.
>
> Bertman et al. start with the following rate equation (Equation 11 in Bertman et al):

$$d[RONO_2]/dt = \beta k_A[RH] - k_B[RONO_2] \qquad \qquad (T1)$$

where $\beta = \alpha_{13}\alpha_{14}$, $k_A = k_{13}[OH]$, $k_B = k_{15}[OH] + j_{16}$  (numbers relate to our manuscript)

N.B. In our manuscript $k_B = k_{15}[OH]$, i.e. we ignore the photolysis sink of the alkyl nitrates.

This assumes a NOx rich environment, i.e. that all peroxy radicals produced react with NO. This will not be the case for an air mass in transport to the Arctic. The peroxy radical could also react with $HO_2$. (This is of course still a simplification - the peroxy radical could also react with other peroxy radicals etc.). To account for this additional sink for the peroxy radical, we simply need to introduce the ratio of the sink of the peroxy radical to [NO] to the total peroxy radical sink. This can be seen by considering the production of $[RONO_2]$ in Equation T1 in terms of $[RO_2]$ (Equation T2).

$$d[RONO_2]/dt = [RO_2][NO]k_{14}\alpha_{14} - k_B[RONO_2] \qquad \qquad (T2)$$

At steady state, which is a valid assumption given the very short lifetime of $RO_2$:

$$[RO_2] = k_A\alpha_{13}[RH] / (k_{14}[NO]+k_{11}[HO_2]) \qquad \qquad (T3)$$

i.e. we now include the loss of $RO_2$ via reaction with $HO_2$ as well as with NO.

 Substituting T3 into T2 gives:

$$d[RONO_2]/dt = \beta k_A[RH]k_{14}[NO] / (k_{14}[NO]+k_{11}[HO_2]) - k_B[RONO_2] \quad (T4)$$

Replacing the ratio $k_{14}[NO]/(k_{14}[NO]+k_{11}[HO_2])$ with $\gamma$ (which assumes the ratio to be a constant):

$$d[RONO_2]/dt = \gamma\beta k_A[RH] - k_B[RONO_2] \qquad \qquad (T5)$$

Note that T5 is exactly the same as Equation 11 in Bertman et al., with the exception of the inclusion of the term $\gamma$. In the same way that Bertman et al. assumed $k_A$ and $k_B$ to be constant (note they are both a function of OH), we also assume $\gamma$ to be constant. We discuss this assumption in a response to a later comment made by the reviewer. The first term on the right hand side of T5 is now [RH] multiplied by a constant equal to $\gamma\beta k_A$ as opposed to being multiplied by a constant equal to $\beta k_A$ in Bertman's equation.

We can then begin to integrate Equation T5. For this we will use integrating factors:

I.e.  if we have an equation of the form:

$$dy/dx + f_1(x)y = f_2(x)$$

then the integrating factor (IF) is:

$$e^{\int f_1(x).dx}$$

and:

$$y = \int IF.f_2(x).dx \; / \; IF$$

Replacing [RH] in Equation T5 with $[RH]_0 e^{-k_A t}$ we get:

$$d[RONO_2]/dt = \gamma \text{\ss} k_A[RH]_0 e^{-k_A t} - k_B[RONO_2] \qquad (T6)$$

For this equation then:

$$f_1(x) = k_B$$

$$f_2(x) = \gamma \text{\ss} k_A[RH]_0.e^{-k_A t}$$

$$IF = e^{\int k_B.dt} = e^{k_B t}$$

Hence:

$$[RONO_2] = \int e^{k_B t}.\gamma \text{\ss} k_A[RH]_0.e^{-k_A t}.dt \; / \; e^{k_B t} \qquad (T7)$$

Solving this gives:

$$[RONO_2] = \gamma \text{\ss} k_A[RH]_0 e^{-k_A t} \; / \; (k_B-k_A) + c.e^{-k_B t} \qquad (T8)$$

At t=0:

$$[RONO_2]_0 = \gamma \text{\ss} k_A[RH]_0 \; / \; (k_B-k_A) + c$$

$$c = [RONO_2]_0 - \gamma \text{\ss} k_A[RH]_0 \; / \; (k_B-k_A)$$

So:

$$[RONO_2] = \gamma \text{\ss} k_A[RH]_0 e^{-k_A t} \; / \; (k_B-k_A) + e^{-k_B t} ([RONO_2]_0 - (\gamma \text{\ss} k_A[RH]_0 \; / \; (k_B-k_A))) \qquad (T9)$$

Replacing $[RH]_0.e^{-k_A t}$ with [RH]:

$$[RONO_2]/[RH] = \gamma \text{\ss} k_A.(1-e^{(k_A-k_B)t}) \; / \; (k_B-k_A) + [RONO_2]_0.e^{(k_A-k_B)t} \; / \; [RH]_0 \qquad (T10)$$

Assuming $[RONO_2]_0 = 0$:

$$[RONO_2]/[RH] = \gamma \text{\ss} k_A.(1-e^{(k_A-k_B)t}) \; / \; (k_B-k_A) \qquad (T11)$$

I.e. Equation E2 from the paper.

Equations T10 and T11 are exactly the same equations as derived by Bertman et al. (Equations 12 and 13), with the exception of the inclusion of the constant $\gamma$. This result can be expected since the constant factor $\text{\ss} k_A$ has simply been replaced by the constant factor $\gamma \text{\ss} k_A$.

We have edited Section 4 somewhat to reflect this discussion, demonstrating that Equation E3 comes from a version of the rate equation for [RONO$_2$]

*As it stands, it is clear that Equation E2 cannot be correct. The left side of the equation is a ratio that is determined by long time scale evolution, while the right side depends on NO and HO$_2$ concentrations, which are variables that vary on very short time scales. For example, if the NO concentration were suddenly increased by fresh emissions, Equation E2 indicates that the [RONO2]/[RH] ratio would change, which is obviously incorrect.*

As stated above, $\gamma$ represents an average of the ratio $k_{14}$[NO]/($k_{14}$[NO]+$k_{11}$[HO$_2$]) over the time t. The air that arrives at the firn drill site will inevitably be a mix of air masses and will have experienced emissions from a number of different source regions. Furthermore, the air extracted from the firn is also a mix of ages and a firn diffusion model is used to derive a time series. As such the [RONO$_2$]/[RH] ratios derived from the firn air represent an integrated effect of the chemical and transport processes. What is really interesting and of value exploring is that these ratios have changed over the period of the firn air record, which suggests that there may have been a change in this rate of processing. In the paper we discuss what these changes might have been. Equations E2 and E3 illustrate that if the air arriving at the firn site in the 1970s had experienced a different mean [NO]/[HO$_2$] to the air in the 1990s, this would affect the observed [RONO$_2$]/[RH] ratios.

*It might be (or might not be) that a proper integration of the appropriate differential equation could yield an equation that looks like Equation E2, where the NO and HO2 concentrations would be weighted time integrals, but those weightings are likely to be different for each species. How those weightings might be calculated is not clear. Equation E3 is derived from Equation E2, so the problems in the latter propagate to the former. The left side is now a ratio of highly variable species, while the right side depends on concentrations that change relatively slowly.*

It is true that changes to the ratio $k_{14}$[NO]/($k_{14}$[NO]+$k_{11}$[HO$_2$]) at different times along the air mass trajectory, t, will affect d[RONO$_2$]/dt at that time differently because d[RONO$_2$]/dt is also driven by [RH] which is a function of time. However, the uncertainties introduced by the assumption of $\gamma$ as a constant on [RONO$_2$]/[RH] calculated at time = 10 days are on the order of 5 % (as demonstrated below), while the observed changes in [RONO$_2$]/[RH] in the firn are considerably larger, on the order of a factor of 4 – 5. Hence we consider the assumption of $\gamma$ as a constant to be a reasonable assumption for the sake of making the problem tractable and that the changes to [NO]/[HO$_2$] that we calculate in the paper are not an artefact of this assumption.

Using Equation T4 (i.e. allowing the ratio $k_{14}[NO]/(k_{14}[NO]+k_{11}[HO_2])$) to vary with time rather than it being a constant as in Equation T5) we can examine the magnitude of changes to [2-butyl nitrate]/[n-butane] in pairs of scenarios where $\gamma$ is the same for each pair but the time evolution of the ratio ($k_{14}[NO]/(k_{14}[NO]+k_{11}[HO_2])$) is different. Figure T1 illustrates how the ratio ($k_{14}[NO]/(k_{14}[NO]+k_{11}[HO_2])$) varies over the 10 days for each of six hypothetical scenarios: A1, A2, B1, B2, C1, and C2. $\gamma$ is the same for both A scenarios and likewise for both B scenarios and both C scenarios. The time evolution of [RH] is calculated as $[RH]_t = [RH]_0 e^{-k[OH]t}$, and [NO] as $[NO]_t = [NO]_0 e^{-t/\tau} + [NO]_{bg}$, where $\tau$ is the NOx lifetime (assumed to be 4 hours (e.g. Liu et al., 2016, ACP)) and $[NO]_{bg}$ is the assumed [NO] in the background atmosphere, and $[NO]_0 = 2.5 \times 10^{11}$ $cm^{-3}$.

The results are presented in Table T1 where we show the percentage difference in $[RONO_2]/[RH]$ at time t = 10 days for each pair of scenarios A, B and C. Scenarios A and B represent 'realistic' scenarios, in that the ($k_{14}[NO]/(k_{14}[NO]+k_{11}[HO_2])$) at time t=0 is ~ 1 and falls to some sort of background value during transport away from an NO source region. This is what can be expected since RH sources are generally co-located with NOx sources (i.e. emissions from motor vehicles) and hence [NO] concentrations are high close to the RH source. Scenario C represents an extreme scenario, but unlikely case, in which $k_{14}[NO]/(k_{14}[NO]+k_{11}[HO_2])$ is 0 for the first half of the transport time and 1 for the second half, or, 1 for the first half of the transport time and 0 for the second half. I.e. $\gamma$ = 0.5 in both cases. Such a scenario is purely hypothetical and serves only to demonstrate the upper range of uncertainty that could arise from the assumption of an average ($k_{14}[NO]/(k_{14}[NO]+k_{11}[HO_2])$). We also consider the results using a range of [OH] in Table T1.

The results in Table T1 show that even though the time evolution of the ratio $k_{14}[NO]/(k_{14}[NO]+k_{11}[HO_2])$ varies, if the average value, i.e. $\gamma$, is the same there is only a small percentage difference between the resulting values for $[RONO_2]/[RH]$. The differences increase with increasing assumed [OH], but even for the largest value of [OH] for Scenarios A and B the greatest difference in $[RONO_2]/[RH]$ at time t = 10 days is < 5%. For Scenario C the difference is up to 35%. However, it is noted that the scenario, in which $k_{14}[NO]/(k_{14}[NO]+k_{11}[HO_2])$ is 0 for the first half of transport and 1 for the second half must be considered extremely unlikely.

[Figure]

**Figure T1** Six hypothetical scenarios with different time evolution of the $k_{14}[NO]/(k_{14}[NO]+k_{11}[HO_2])$ ratio but where $\gamma$ is the same for both A scenarios and likewise the same for both B and C scenarios.

**Table T1** Percentage difference in $[RONO_2]/[RH]$ after 10 days between the scenarios within each pair shown in Figure T1 for $[OH] = 2, 4$ and $6 \times 10^5$ cm$^{-3}$. $[RONO_2]/[RH]$ calculated using Equation T5. $[RH]$ calculated as $[RH]_t = [RH]_0 e^{-k[OH]t}$.

| [OH] / cm$^{-3}$ | $2\times10^5$ | $4\times10^5$ | $6\times10^5$ |
|---|---|---|---|
| Scenarios A ($\gamma$=0.36) | 1.2 | 2.5 | 4.8 |
| Scenarios B ($\gamma$=0.77) | 1.3 | 2.4 | 3.4 |
| Scenarios C ($\gamma$=0.5) | 14.0 | 24.7 | 34.7 |

*The authors are evidently implicitly assuming that the NO and HO2 concentrations are some sort of remote, hemisphere wide average (as indicated by the statement on lines 11-12 of page 13). However, this is far from clearly correct, as the production of the alkyl nitrates may well be dominated by relatively rapid photochemistry in the urban, NOx-rich, higher OH environment, rather than photochemistry in the remote environment. Hence, properly integrating the differential equations describing the photochemical processes is critical to correct analysis. That integration may not be possible to carry out analytically.*

In urban areas, close to the RH source, there is no scope for a change in the production efficiency of alkyl nitrates because $(k_{14}[NO]/(k_{14}[NO]+k_{11}[HO_2])) \sim 1$ (i.e. production is RH limited). In the rural / background environment, where $(k_{14}[NO]/(k_{14}[NO]+k_{11}[HO_2])) < 1$, changes to [NO] would have an effect on the alkyl nitrate production efficiency.

That being said, an increase in urban NOx would have a small effect in terms of $(k_{14}[NO]/(k_{14}[NO]+k_{11}[HO_2]))$ remaining at ~1 for a longer time during air mass

transcript. Using Scenarios A1 and B1 (above), an increase of $[NO]_0$ of two orders of magnitude (i.e. to $2.5 \times 10^{13}$ cm$^{-3}$) yields changes in $[RONO_2]/[RH]$ at time t = 10 days of 18% for Scenario A1 and 6% for Scenario B1, assuming $[OH] = 6 \times 10^5$ cm$^{-3}$.

Figure T2 shows the time evolution of $[RH]$ (in terms of % remaining compared to $[RH]$ at t=0), % of total $RO_2$ produced, and $(k_{14}[NO]/(k_{14}[NO]+k_{11}[HO_2]))$, for Scenario A1 above ($[NO]_0 = 2.5 \times 10^{11}$ cm$^{-3}$ (i.e. 10 ppb), the NOx lifetime in the air mass = 4 hours, background $[NO] = 1.9 \times 10^7$ cm$^{-3}$, $[HO_2] = 2 \times 10^7$ cm$^{-3}$, initial $[RH] = 800$ ppt, $[OH] = 6 \times 10^5$ cm$^{-3}$). It is seen that the vast majority (i.e. 86%) of the $RO_2$ produced over the 10 days is produced when $(k_{14}[NO]/(k_{14}[NO]+k_{11}[HO_2])) < 1$. (If $[OH]$ were lower than $6 \times 10^5$ cm$^{-3}$ then the amount of $RO_2$ produced when $(k_{14}[NO]/(k_{14}[NO]+k_{11}[HO_2])) < 1$ would be even larger.)

This shows that there is a large scope for changes in the ratio $(k_{14}[NO]/(k_{14}[NO]+k_{11}[HO_2]))$ away from sources to affect alkyl nitrate production efficiency during air mass transport. Figure T3 shows the time evolution of the $[RONO_2]/[RH]$ ratio for the scenario shown in Figure T2. It also shows the $[RONO_2]/[RH]$ calculated assuming $(k_{14}[NO]/(k_{14}[NO]+k_{11}[HO_2]))$ to remain at 1 for the whole transport time. Figure T2 shows that for the scenarios shown, an increase in $\gamma$ has the potential to increase $[RONO_2]/[RH]$ at t = 10 days by a factor of 2.5.

[Figure]

**Figure T2** The time evolution of $[RH]$, % of total $RO_2$ produced, and $(k_{14}[NO]/(k_{14}[NO]+k_{11}[HO_2]))$, calculated using $[NO]_0 = 2.5 \times 10^{11}$ cm$^{-3}$, NOx lifetime = 4 hours, and $[OH] = 6 \times 10^5$ cm$^{-3}$, background $[NO] = 1.9 \times 10^7$ cm$^{-3}$, and $[HO_2] = 2 \times 10^7$ cm$^{-3}$. The vertical dashed line represents the time when $k_{14}[NO]/(k_{14}[NO]+k_{11}[HO_2]$ drops below 0.98, and the horizontal line indicates that only 14% of the $RO_2$ is produced when $k_{14}[NO]/(k_{14}[NO]+k_{11}[HO_2]$ is close to 1.

[Figure]

**Figure T3** The time evolution of $[RONO_2]/[RH]$ for the scenario shown in Figure T2. Solid blue line shows $(k_{14}[NO]/(k_{14}[NO]+k_{11}[HO_2]))$ from Scenario A1, dashed blue line shows maximum possible $(k_{14}[NO]/(k_{14}[NO]+k_{11}[HO_2]))$, i.e. equal to 1 for the whole transport time. Solid purple line shows $[RONO_2]/[RH]$ calculated using $(k_{14}[NO]/(k_{14}[NO]+k_{11}[HO_2]))$ from Scenario A1 ($[RONO_2]$ calculated with Equation T4, $[RH]$ calculated as $[RH]_t = [RH]_0 e^{-k[OH]t}$, as above). Dashed green line shows $[RONO_2]/[RH]$ calculated using $(k_{14}[NO]/(k_{14}[NO]+k_{11}[HO_2])) = 1$ for the whole run.

In summary, in our response to Referee #1's point 1, we have demonstrated that the $[RONO_2]/[RH]$ ratios seen at the firn air site can be influenced by the $[NO]/[HO_2]$ ratios during transport. Thus changes to the background concentrations of NO or changes to its lifetime that lead to an overall change in the average value of $k_{14}[NO]/(k_{14}[NO]+k_{11}[HO_2]$ can result in changes in the ratios in $[RONO_2]/[RH]$ at remote locations. We accept that there are large assumptions in the methodology in our paper, but we believe that the assumptions we have made are clear, justified and consistent with other assumptions inherent in Bertman et al.'s original equations and therefore are appropriate for interpreting the firn air data.

The nature of these data means that they represent averages of air masses arriving at the firn site from different regions and over extended periods of time. We use simplified equations to illustrate that changes in the $[NO]/[HO_2]$ ratios could contribute to the decadal changes in the firn $[RONO_2]/[RH]$ data. We believe this is an important point to make and adds to the growing evidence for changes in atmospheric chemical processing during this period, but we acknowledge that this needs to be further fully investigated through the use of more sophisticated models that can then take account of the complexities of the chemistry and transport.

***2) It is unclear whether the uncertainties in the estimated diffusion coefficients of the alkyl nitrates have an important impact on the atmospheric history reconstructions. This should be discussed.***

Author response:

Figure T4 shows the three alkyl nitrate scenarios presented in Figure 2 in the paper. A red dashed line is included using diffusion coefficients calculated using the method of Chen and Othmer (1962) which gives diffusion coefficients some 10% lower than those calculated using the method of Fuller et al. (1966) which are used for the work presented in the manuscript. It is seen in Figure T4 that the difference between the scenarios derived using the two differently calculated diffusion coefficients are minor and well within the uncertainty envelopes.

[Figure]

**Figure T4** As Figure 2 in manuscript but including red dashed line showing effect of 10% decrease in diffusion coefficient used in firn modelling.

We have included the following additional text in the third paragraph of Section 2.3 (Firn Modelling):

"*Model runs were also performed using diffusion coefficients for the alkyl nitrates calculated using the Chen and Othmer method. These coefficients are ~ 10% lower than those calculated using the Fuller method. However, the atmospheric scenarios derived from the modelling are very similar, well within the 2-$\sigma$ uncertainty envelopes presented in Figure 2.*"

**3) On page 7 "It is noted that the latter part (post-1995) of the model derived scenarios for 2+3-pentyl nitrate is rather sensitive to the inclusion or exclusion of the measurement at 34.72 m (the most shallow measurement used). The scenarios presented in this work are based on including this measurement." Is this an important issue? The authors should discuss the effect on the final conclusions if this measurement is excluded.**

There is not a strong argument for removing the data point, however we felt the need to make clear that there has been some consideration as to the possible effect of seasonality on the alkyl nitrate measurements at this depth. In Helmig et al. 2012, a series of tests were performed using the firn model to examine the effect of seasonality on the alkane measurements at the NEEM sites between 30 and 40 m depth (Figure 6, Helmig et al. 2014). The alkanes have a similar diffusion coefficient, and similar atmospheric seasonality, to the alkyl nitrates and so any seasonality effect would be very similar. It is seen in this figure that the effect at 35 m for the butanes and pentanes (the most similar alkane molecules to the alkyl nitrates presented here) is < 5 %.

If the measurement were affected by seasonality, the correct thing to do would be to apply a minor correction rather than to remove the point. These two processes would have very different effects. Removing the measurement has a significant effect on the derived reconstruction because of the limited size of the data set in that region. Correcting for any seasonality on the other hand would have a very minor effect on the derived reconstruction.

As stated in the paper, removing the measurement at 34.72 m would have a significant effect on the derived scenarios post-1995, particularly for 2+3-pentyl nitrate. This is because we are removing a constraint on a data set of limited size - the decreasing concentrations post-1995 for the two pentyl nitrates are constrained by relatively few measurements partly because of the increased scatter in the peak region (55 - 65 m) compared to 2-butyl nitrate. However, this is already demonstrated in our data presentation by the relative sizes of the 2-sigma uncertainty envelopes for the pentyl nitrates compare to 2-butyl nitrate (~ 10% at the peak for 2-butyl nitrate compared to 25 - 30% for the pentyl nitrates).

*Minor issues:*

*1) Abstract lines 18-21 - I disagree with the statement "Due to their short atmospheric lifetimes, NOx and HOx are highly variable in space and time, and so the measurements of these species are of very limited value for examining long term, large scale changes to their budgets." If measurements of either HOx or NOx were measured in a given region in an extensive enough manner to characterize average ambient concentrations (e.g., during an intensive aircraft deployment such as NASA DC-8 studies), and if those extensive measurements were repeated after the passage of a decade or so, then it should be possible to quantify the long term, large scale changes to their concentrations. I suggest that this statement be removed, as it is superfluous to the paper.*

Author response:

We have removed the word 'very' from the statement highlighted. However, we maintain the essence of the statement to be correct.

The study that the referee suggests would, in their words, require measurements that are in an "extensive enough manner" and if these were repeated then it would be possible to make an estimate of a long term trend. However, such a study requires such a large spatial and temporal coverage to be able to give meaningful results as to make it either prohibitively expensive or an extremely rare experiment. By far the majority of the measurements of NOx and HOx are not like this and so we stand by our statement that "measurements of these species are of limited value for examining long term, large scale changes to their budgets". We believe that this is a justifiable reason for alternative methods of examining the problem. This is entirely consistent with the rationale of using methyl chloroform for determining long term changes in OH.

*2) Page 2, lines 13 and 14. I also disagree with the phrase "which is positively correlated with NOx concentrations in the background atmosphere through the photolysis of NO2 (Reactions R1-R2)." Ozone is certainly correlated with total NOy concentrations, but since ozone and NOx have very different lifetimes, they are in general only poorly correlated, because any correlation resulting from production through the photolysis of NO2 is destroyed by removal or fresh emissions of NOx, even in the remote troposphere. Here and elsewhere throughout the paper, I suggest that very careful consideration be given to the veracity of each sentence.*

Author response:

Our intended meaning was that "ozone production" was positively correlated to NOx concentrations. We see that our statement was ambiguous so we will change the

sentence to "$NO_X$ and $HO_X$ are linked through ozone production, which is positively correlated with $NO_X$ concentrations in the background atmosphere through the photolysis of $NO_2$ (Reactions R1-R2)."

**3) Page 2, lines 23 and 24 - The phrase "The main removal processes for HOx are the reaction of OH with NO2 (Reaction R9) ..." has an error. Evidently this reaction has not been separately listed in the paper.**

Author response:

The OH + NO2 reaction is already included but the referee is correct it is not Reaction R9, it is Reaction R5. This has been changed.

**4) Page 4, line - Should be "on-road" vehicles.**

Author response:

This has been changed to read '…from motor vehicles.'.

**5) Page 6 - Sentence on lines 18-21 is not clear.**

Author response:

The second 'samples' has been removed from line 19. The sentence hopefully now makes sense.

**6) Page 8 - Line 12 - The Worton et al. (2010) reference is not in the References list; should be Worton et al. (2012)?**

Author response:

The referee is correct. Changed.

**7) Page 12, lines 19-22 are not exactly correct: In Equation E1, taken from Bertman et al. (1995), [OH] is assumed to be the average over the time t. This is exactly what the authors are doing as well. This should be clarified, i.e. the term [OH]t represents the time integral of [OH] over the period during transport from the source region to the Arctic.**

Author response:

Lines changed to:

"In Equation E1, taken from Bertman et al. (1995), [OH] is assumed to be a constant. Similarly for the purposes of this work, [OH] is assumed to be constant and represents an average [OH] to which the air mass is exposed during transport from the source region to the Arctic."

---

## Editor Decision (ED1)

I am in a bit of a conundrum here.  One of the reviewers has suggested rejecting this paper.  The second reviewer has suggested publishing the paper "as is", to essentially let the chips fall where they may.  I agree with the one reviewer who suggested that the alkyl nitrate firn measurements should be published, however the second half of the publication is by no means without uncertainly (several).  I agree that with the current uncertainties in the paper, it difficult to see convincing evidence to make conclusion that the chemical state of the northern hemisphere has changed.  I find many of the arguments in the current version of the paper to be uncertain, with many assumptions untested that raise many questions.  I think that this is a potentially very important paper but if it is to be published, changes will still need to be made to address the uncertainties that still exist. If you can address these concerns, especially my major uncertainty (see below), then send a revised paper to me to consider.

I have several small concerns, some that are shared with the reviewer(s) and one major uncertainty that has not been raised by either reviewer.

Robert McLaren
Co-editor ACP

**Major uncertainty**

It is well know that the major sources of the parent compounds (iC4, nC4, iC5 alkanes as well as n-pentane) derive from fuel and fuel evaporation sources.  These species are among the most volatile in gasoline products and as such they make up a major fraction of the vapor pressure of gasoline (or at least historically they did), typically expressed as the Reid Vapor Pressure (RVP).  It is also well known that prior to late 80's/early 90's the % composition of gasoline was very high in these C4/C5 components due to carburetor's being very common and that refineries typically adjusted the RVP of gasoline according to the season (high in winter to help cold start; low in summer to avoid vapor lock) and according to the latitude of the market; where one major parameter that refineries used to adjust the RVP ***was the % composition of C4 and C5 components of the gasoline***.  Thus there were major changes to the seasonal composition of the major source of C4 and C5 alkanes to the atmosphere; this was far from being constant seasonally in the 70's, 80's, 90's.  Furthermore, in the 90's when vapor regulations started to come in to affect in a big way to control the tropospheric O3 problem, especially in North America (presumably Europe as well, I am not clear on this) the regulations dictated that RVP of gasoline be reduced significantly (down to RVP = 6 or 7 psi in California as I recall).  As far as I understand there were major reductions in the C4/C5 components in gasoline at this time.  Thus, sources of C4/C5 emissions changed significantly, and likely they changed both in magnitude and the spatiotemporal patterns changed (southern areas reduced RVP more than northern areas, California was first).   This could conceivably have changed not just the sources of C4/C5 alkanes to the atmosphere but also the temporal transport times to the arctic.

I see virtually no discussion of the points raised above, including major reformulations in gasoline that have taken place historically due to changing technology (Carburetor to Fuel injection vehicles) and changing regulations (reduced RVP and increased vapor recovery in southern locations), other than page 4, line 26/27, which says that *"Emissions are not thought to have a significant seasonal variability*

*(Pozzer et al, 2010)"*.  While this statement may currently be the case, it certainly was not the case in the past and it does not address the temporal changes in emissions over decades from 1970 onward, and any potential spatio-temporal change in the emissions pattern that resulted from changing regulations. Thus not only [RH] changes seasonally and over decadal periods, but the ratio [RONO2]/[RH] could change over decades also due to changing emissions and emission patterns.  Thus changes in [RONO2]/[RH] could partially be driven by changes in emissions. Likewise changes in $\Delta$[NO]/[HO2] and $\Delta$[OH]t (Figures 4 and 6) could partially arise from changes in emissions as well.  While what I lay out above is somewhat speculative (changing spatiotemporal patterns), the changes in emissions, reformulations and changes in regulations that affect C4/C5 precursor hydrocarbons are not.  I think this should be addressed before a conclusion is made that the chemical state of the atmosphere has changed.

**Other Major Concerns and Uncertainties**

P12L20 – what is the effect in this term of ignoring RO2-RO2 self reactions.

P13L1: it is not good enough to just say that we assume the $\gamma$ is constant, since we know that $\gamma$ is not constant and that it changes as the air mass transports from source regions to the arctic.  You must justify the statement by telling us what the estimated effect is …perhaps by telling us in a few lines what was present in Table T1 (your response to reviewer.)   Table T1 : Your scenarios should probably have extended [OH]  at least to the generally accepted day and night global average [OH] level, [OH] = $1.2 \times 10^6$ molec cm$^{-3}$, as a sensitivity test.  Obviously the changes in [RONO$_2$]/[RH] in this case would be larger than 5% for both A and B scenarios.  The uncertainties you discuss in this response to the reviewer should be added to the paper as a few lines and perhaps to Supplemental.

P13 L28: Equation 13.  [NO]/[HO$_2$] should be presented with some sort of average symbol that is different from what you have now, which implies an instantaneous average of concentrations instead of a long term average.  ie [NO]~ $\int$ [NO]dt or [NO]/[HO$_2$] = $\int$ [NO]/[HO$_2$] dt

E3: The [NO]/[HO$_2$] average you calculate and present in Figure 4 depends (perhaps critically so) on the values of all rate constants and branching ratios in the equations.  The constants assumed should be presented somewhere in the paper and/or you should tell us at what temperature you calculated the rate constants (since T likely changes 30-40 $^o$C as the air mass is transported from source regions to the arctic the region).  What is the uncertainty in assuming a constant T.  What is sensitivity of Figure 4 to using different T?

Figure 14L4 (Figure 4 caption).  Why is photochemical age $\int$ [OH]dt = 5 x10$^{11}$??  This is about 5 days averaging at global average [OH].  How does figure change as you change the photochemical processing?

p17E6 – I believe this equation is derived presuming that alkyl nitrates do not photolyze.  Considering that ~ 50% of loss of 2-butyl nitrate is photolysis, what is the effect on the calculations and the results in Figure 6 resulting from this assumption.  Also why is [NO]/[HO$_2$] presumed to be 0.5.  Reference? rationale?  Sensitivity?

**Minor Points**

p2L23-24: by removal, are you presuming CH4 +OH and CO + OH regenerate OH catalytically. In any case, please provide a reference for such statements.

page 2 - HOx Sources: In winter regions, O3 photolysis may not be the main source of OH. There is lots of recent evidence for this.

Page 2-3 and throughout: The chemical equations are not presented with care. Three body reactions should be presented as such (R2 for example). Photolysis reactions should be indicated as photolysis reactions. (R1, R3, etc). R7 is an equilibrium reaction. Reaction 13 is not balanced, namely because it is actually 2 reactions. R14a does not exclusively give aldehydes as shown, it also give ketones (ie- MEK from butane). R14b involves a rearrangement that is apparently highly temperature dependent that is not mentioned.

p4L26 – Emissions definitely did have seasonal patterns in the past. I am not sure about today.

p7L27 – Your answer to the question by a reviewer about the effect of the 34.7m sample was not convincing. If this sample was removed, do your conclusions change; regardless of whether it should be removed or not. The historical state of the atmosphere should not rest on a single point.

Figure 3 – how many points are represented in the curve for n-butane in this figure. How uncertain is the peak year, or put another way, what is the uncertainty in the difference between the peak year for n-butane and 2-butyl nitrate. I presume that the shape of Figures 4 and Figures 6 depend critically on the relative temporal trend of n-butane measured in the firn from the other paper. Does the shape of n-butane in Figure 3 make sense given the temporal changes in hemispheric emissions of n-butane…can it be corroborated with other sources of hydrocarbon measurements in northern hemisphere cities or other sites??

p11L4. At least once in this paper you should acknowledge that [OH]t is not a constant, it represents $\int$ [OH] dt, and perhaps it would better be presented as an average symbol.

p13L15 – [NO] can range from 1ppb to ??? I have never seen the upper end of your range, 1000 ppb?? Provide a reference if so.

P15 Figure 5 –for comparison, both y-axis should extended to zero. Units missing on right axis.

p16L20 Section 5.2. Does 1 sentence deserve its own section?

p20L12-15: other primary sources of O3. HCHO? HONO??

---

## Author Response (AR2)

Dear Professor McClaren,

Firstly, thank you for taking the time to closely consider the reviewers' comments and to thoroughly review the paper yourself. We appreciate that the reviewers' contrasting comments puts you in a difficult position so we are grateful that you have given us this further opportunity to improve the quality and clarity of the paper.

We of course agree with the reviewer who suggested that the paper should be published, allowing others to read it, include our new data in their own research and to consider our analysis of that data. The paper presents measurements which appears to shed light on an issue that is absolutely central to the community's understanding of historical trends in the chemical composition of the troposphere. Only in the last few weeks, two other papers have been published in PNAS on this subject (Rigby et al., Role of atmospheric oxidation in recent methane growth; Turner et al., Ambiguity in the causes for decadal trends in atmospheric methane and hydroxyl). In those papers they used atmospheric measurements of methyl chloroform and $\delta^{13}CH_4$ to derive a trend for OH. Their conclusion was that a change in the oxidising capacity could be partially responsible for the accelerated growth in methane post 2007, but they too had to accept that no change in OH was still consistent with their results given the large uncertainties. The trend in oxidizing capacity of the atmosphere is proving difficult to determine given all the uncertainties in measurements, emission inventories and modelling techniques. However, we firmly believe that our data and analysis provide a new angle to this debate and, whilst it too comes with large uncertainties which we openly acknowledge, we think it is important to get published so that it can contribute to solving this important issue.

To summarise how we see what we have done in the paper: we present a new data set (alkyl nitrate trends), and then ask why the trends look as they do. We combine the alkyl nitrate trends with existing data sets (parent alkanes) to gain more insight, and then present a range of possible explanations that could explain the observed trends. We are clear where assumptions have had to be made. Neither reviewer presents any alternative explanations for the observed trends beyond those we posit. As we see it, this is everything that should be done in a paper. That the paper may cause discussion and debate over its conclusions can only be seen as a good thing, hopefully leading to further work in the area and ultimately improving our understanding of tropospheric chemistry.

I note that the title, abstract and conclusions taken together were somewhat inconsistent. The title and the conclusions explicitly suggested that the observations presented in the paper were driven by changes to the chemical state of the atmosphere whereas the abstract presents all possibilities equally. The conclusions have been altered to give a more equal weight to all possibilities.

We hope that you will feel that we have addressed your comments satisfactorily and will agree to publishing our manuscript.

Below we respond to each of your comments (in bold) in turn. Our comments are inset. Altered text from the manuscript is inset and italicised. We then refer to specific lines from this section in our responses.

**Major uncertainty**

It is well know that the major sources of the parent compounds (iC4, nC4, iC5 alkanes as well as n- pentane) derive from fuel and fuel evaporation sources. These species are among the most volatile in gasoline products and as such they make up a major fraction of the vapor pressure of gasoline (or at least historically they did), typically expressed as the Reid Vapor Pressure (RVP). It is also well known that prior to late 80's/early 90's the % composition of gasoline was very high in these C4/C5 components due to carburetor's being very common and that refineries typically adjusted the RVP of gasoline according to the season (high in winter to help cold start; low in summer to avoid vapor lock) and according to the latitude of the market; where one major parameter that refineries used to adjust the RVP *was the % composition of C4 and C5 components of the gasoline*. Thus there were major changes to the seasonal composition of the major source of C4 and C5 alkanes to the atmosphere; this was far from being constant seasonally in the 70's, 80's, 90's. Furthermore, in the 90's when vapor regulations started to come in to affect in a big way to control the tropospheric O3 problem, especially in North America (presumably Europe as well, I am not clear on this) the regulations dictated that RVP of gasoline be reduced significantly (down to RVP = 6 or 7 psi in California as I recall). As far as I understand there were major reductions in the C4/C5 components in gasoline at this time. Thus, sources of C4/C5 emissions changed significantly, and likely they changed both in magnitude and the spatiotemporal patterns changed (southern areas reduced RVP more than northern areas, California was first). This could conceivably have changed not just the sources of C4/C5 alkanes to the atmosphere but also the temporal transport times to the arctic.

I see virtually no discussion of the points raised above, including major reformulations in gasoline that have taken place historically due to changing technology (Carburetor to Fuel injection vehicles) and changing regulations (reduced RVP and increased vapor recovery in southern locations), other than page 4, line 26/27, which says that *"Emissions are not thought to have a significant seasonal variability (Pozzer et al, 2010)".* While this statement may currently be the case, it certainly was not the case in the past and it does not address the temporal changes in emissions over decades from 1970 onward, and any potential spatio-temporal change in the emissions pattern that resulted from changing regulations. Thus not only [RH] changes seasonally and over decadal periods, but the ratio [RONO2]/[RH] could change over decades also due to changing emissions and emission patterns. Thus changes in [RONO2]/[RH] could partially be driven by changes in emissions. Likewise changes in Δ[NO]/[HO2] and Δ[OH]t (Figures 4 and 6) could partially arise from changes in emissions as well. While what I lay out above is somewhat speculative (changing spatiotemporal patterns), the changes in emissions, reformulations and changes in regulations that affect C4/C5 precursor hydrocarbons are not. I think this should be addressed before a conclusion is made that the chemical state of the atmosphere has changed.

> We think that much of this concern has derived from our inclusion of the line, *"Emissions are not thought to have a significant seasonal variability (Pozzer et al, 2010)".* We agree that we have no evidence of this beyond citing Pozzer et al. and will remove the line. However, any seasonal variability of emissions driven by changes in

fuel composition in US cities is of little relevance to the Arctic measurements presented, hence the reason we haven't discussed it in the paper.  We shall explain below:

(i)     The observed alkyl nitrate and alkane signals captured in the Arctic firn are caused almost entirely by winter time (October-May) emissions (alkanes) and chemical production (alkyl nitrates). This is seen in the seasonal cycles of the alkanes and alkyl nitrates at Summit (Swanson et al., 2003) where the winter maxima dominate over the summer minima, because of the short lifetimes of the alkanes and alkyl nitrates in the summer. Whereas, *"EPA regulates the vapor pressure of gasoline sold at retail stations during the summer ozone season (June 1 to September 15)"* (https://www.epa.gov/gasoline-standards/gasoline-reid-vapor-pressure). Hence this summer time mix of alkanes will have minimal impact on the smooth Arctic signal.

To demonstrate this point further, the Arctic records of the C4 and C5 alkanes presented in Helmig et al. (2014) show very similar trends. I believe that the main summer-time change to US fuel composition was to greatly reduce n-butane content (rather than iso-butane, n-pentane or iso-pentane) see e.g. Gentner et al., 2009, ES&T. However, there is no change in the Arctic n-butane record relative to the other alkanes at the time of the implementation of these measures (early 1990s).

Further to all of this is that numerous studies have suggested that Europe is the main winter time contributor to Arctic pollutants and as far as I am aware, no similar policy to that in the US has ever been implemented in European countries.

Further again, as stated above, this US legislation only comes in from 1990 (in California where it was introduced first). We see most of the increases in alkyl nitrate mixing ratios and changes in ratios with their parent alkanes before any of these measures were introduced.

So while we propose to put in a few lines clarifying why these well-known seasonal variations to fuel composition in US cities have no bearing on our work, I hope that the editor sees why we do not consider that the issue requires a major discussion in the paper.

In the Introduction (Section 1.1), we will add the lines,

*"Butane and pentane emissions will be dependent on fuel composition, with evaporative emissions also dependent on temperature. Many areas in the United States are part of 'ozone attainment areas' and during summer months (June-September 15) have been required by law since 1989 to provide a gasoline blend with a low Reid vapour pressure (RVP) to reduce the ozone production potential (www.epa.gov). This reduction in RVP is generally achieved by reducing the fuel's butane content relative to winter-time fuel (e.g.*

*Gentner et al., 2009). Measurements in firn air from Greenland (Aydin et al., 2011; Worton et al., 2012; Helmig et al., 2014) suggest northern hemisphere C2-C5 alkane mixing ratios increased through the middle of the past century to a peak in ~1980 (~1970 for ethane). In-situ measurements from the urban areas of London (1993 – 2008) (Dollard et al., 2007; von Schneidemesser et al., 2010) and Los Angeles (1960 – 2010) (Warneke et al., 2012) show steadily decreasing alkane mixing ratios, as do measurements at the semi-rural site of Hohenpeissenberg, Germany (von Schneidemesser et al., 2010). Emission estimates from the ACCMIP global emission inventory (Lamarque et al., 2010) (available at http://eccad.sedoo.fr) show non-methane VOC emissions in Europe and North America increasing steadily between 1950 and 1980 before falling again to roughly 1965 levels by 2000."*

(ii)    The great benefit of the method of analysis presented in this paper, looking at the ratio of the alkyl nitrates to their parent alkanes, is that it is not affected by long term changes to alkane emissions driven by changes in technology etc., unless these changes happened at different times in different countries / states. i.e. if these changes significantly alter the transport/processing time between the source region and the Arctic.

The point that fuel compositions may have been changed at different times in different locations is of course relevant. Changing the source regions of the alkanes will affect the amount of processing time between the source region and the Arctic (i.e. $\overline{[OH]}t$) (as we discuss in Section 6.1).

Another driver of this sort of spatial change could have been either of the points that you raise, i.e. inter-annual (rather than intra-annual) changes to fuel composition driven either by technology or regulation regulation in different places (i.e. European countries / US states) at different times could have had an effect on spatial distribution of emissions. However there is no evidence for this, with estimates of butane and pentane emissions from ACCMIP (http://eccad.sedoo.fr) for the two regions displaying very similar trends.

Furthermore, my understanding is that RVP was largely controlled in the US by changes to butanes specifically (not pentanes). The fact that we are seeing a very similar trend for the butanes compared to the pentanes again suggests that summer-time fuel composition changes are having little/no effect on the signal we see in the Arctic.

So while these are all indeed possibilities, there seems little evidence for them. However, we now include a discussion on these possibilities in Section 6.1 with more possible reasons behind changes to atmospheric transport time/source regions.

*"Concerning (ii), changes to the relative distribution of the major source regions of the alkanes could have occurred for a number of reasons. Fuel composition has changed through time as a response to technological development of vehicles. Clean air legislation has led to the development of cars with progressively lower evaporative and tailpipe emissions (e.g. Wallington et al., 2006), through developments such as catalytic converters. In addition emissions may have changed simply due to a change in vehicle usage. If such changes were to have occurred in more northerly regions significantly earlier than in more southerly regions, this could have increased the mean transport time of air masses to the Arctic.*

*For many areas in the United States, the Reid vapour pressure of fuel is regulated in the summer season (June 1 - September 15) (epa.gov.uk), leading to sale of a different fuel mix in summer compared to winter. This is generally achieved by producers reducing the butane content of the fuel (Gentner et al., 2006). This legislation came in in 1990. However, the observed alkane and alkyl nitrate signals in Greenland are almost entirely winter-time signals (e.g. Swanson et al., 2003), and so such seasonal variation in fuel composition would not be expected affect the firn measurements.*

*The main sources of anthropogenic emissions to the Arctic of gases with lifetimes on the order of a few weeks, particularly during the winter, have been identified as being northern Eurasia (e.g. Shindell et al., 2008; Stohl et al., 2007; Klonecki et al., 2003). Emissions from Europe and North America have followed a similar declining trend in recent years (Lamarque et al., 2010; von Schneidemesser et al., 2010; Warneke et al., 2012;), thus the relative contribution from each region will not have changed dramatically."*

To summarise our response to your major concern, we don't feel that seasonal changes to fuel composition (and thus emissions) in the US, will have any effect on our results. We agree that changes to winter-time emissions, driven by technology or legislation, at different times in different countries in Europe or states in the US (or Canada) could change the atmospheric processing time of air masses arriving in the Arctic, but have found no evidence of such changes.

**Other Major Concerns and Uncertainties**

**P12L20 – what is the effect in this term of ignoring RO2-RO2 self reactions.**

Effectively the production efficiency term, which we now represent explicitly in Equations E4, E5 and E6, represents $k_{14}[NO]$/total sinks, where total sinks = $k_{14}[NO]+k_{11}[HO_2]+\Sigma k[RO_2]$+etc.

However, in the background environment, the only other peroxy radical possibly present at high enough concentrations to compete with NO and $HO_2$ as a sink is $CH_3O_2$, which is thought to be present at similar concentrations to $HO_2$. However, the reaction

rate of $CH_3O_2$ with other alkyl peroxy radicals is < $2x10^{-13}$ $cm^{-3}$ $s^{-1}$ (http://iupac.pole-ether.fr/htdocs/show_datasheets.php?category=Gas-phase+organics%3A+ROO), i.e. two orders of magnitude slower than Butyl peroxy + $HO_2$ ($2.1x10^{-11}$ $cm^{-3}$ $s^{-1}$, MCMv3.3.1). In an urban environment the term $k_{14}[NO]$ would be expected to dominate. Hence we present $\gamma$ simply as $k_{14}[NO]/k_{14}[NO]+k_{11}[HO_2]$.

We have added comments to this effect to Section 4:

"*We extend Equation E2 to include the possibility of alkyl nitrate production at less than 100% efficiency, in a non-NOx-rich environment, i.e. that the peroxy radical, $RO_2$, formed may react with something other than NO. This is achieved by the inclusion of the term $k_{14}[NO]/(k_{14}[NO]+other\ RO_2\ sinks)$. In high-NOx environments, this value is $\cong$ 1. However, in lower NOx environments, other sinks for the peroxy radical, $RO_2$, will compete with NO. In reality the term $k_{11}[HO_2]$ is likely to dominate the 'other $RO_2$ sinks' term in a background environment, with the only other species likely to be present at high enough concentrations to compete, being the methyl peroxy radical ($CH_3O_2$), which may be present at similar concentrations to $HO_2$, but the reaction rate of $CH_3O_2$ with other alkyl peroxy radicals larger than $CH_3O_2$ is $\leq 2\times10^{-13}$ $cm^{-3}$ $s^{-1}$ (IUPAC), two orders of magnitude slower than the reaction with $HO_2$ (IUPAC). Hence in Equation E3 we extend Equation E2 by including the term $k_{14}[NO]/(k_{14}[NO]+k_{11}[HO_2])$.*"

**P13L1: it is not good enough to just say that we assume the γ is constant, since we know that γ is not constant and that it changes as the air mass transports from source regions to the arctic. You must justify the statement by telling us what the estimated effect is ...perhaps by telling us in a few lines what was present in Table T1 (your response to reviewer.)**

We have added some of the comment to Reviewer #1 into Section 4 (see below). and have included most of the comment with some plots in Supplementary Information. The values in Table S1/T1 have changed (decreased) as we now include the photolysis sink as throughout the rest of the paper.

"*For the purposes of our calculations, $k_{14}[NO]/(k_{14}[NO]+k_{11}[HO_2])$ is assumed (in the same way as [OH]) to represent an integrated value for this ratio during transport of the air mass from the source region to the Arctic, i.e. 1/t * $\int k_{14}[NO]/(k_{14}[NO]+k_{11}[HO_2]).dt$. This is represented by the term $\gamma$ in Equation E4.*

$$\gamma = \overline{\left(\frac{k_{14}[NO]}{k_{14}[NO]+k_{11}[HO_2]}\right)} = Mean\ alkyl\ nitrate\ production\ efficiency$$

*$k_{14}[NO]/(k_{14}[NO]+k_{11}[HO_2])$ would not be expected to be constant in reality since [NO] is likely to change by orders of magnitude during transport, with values on the order of $2.5\times10^{11}$ $cm^{-3}$ close to the emissions source falling to ~$1\times10^8$ $cm^{-3}$ further from source. However, while changes to the ratio $k_{14}[NO]/(k_{14}[NO]+k_{11}[HO_2])$ at different times along the air mass trajectory will affect $d[RONO_2]/dt$ at that time differently because $d[RONO_2]/dt$ is also driven by [RH] which is a function of time, the uncertainties introduced by the assumption of $\gamma$ as an integrated value on*

*[RONO₂]/[RH] calculated at time t = 10 days are on the order of 5 % (see Supplementary Information). The observed changes in [RONO₂]/[RH] in the firn are considerably larger than this, on the order of a factor of 3 – 5. Hence we consider the assumption of γ as a constant to be a reasonable assumption for the sake of making the problem tractable and that the changes to γ that we calculate in the paper are not an artefact of this assumption."*

**Table T1 : Your scenarios should probably have extended [OH] at least to the generally accepted day and night global average [OH] level, [OH] = 1.2x106 molec cm-3, as a sensitivity test. Obviously the changes in [RONO2]/[RH] in this case would be larger than 5% for both A and B scenarios. The uncertainties you discuss in this response to the reviewer should be added to the paper as a few lines and perhaps to Supplemental.**

The air which we are sampling has not been exposed to global mean [OH], it has been exposed to the winter-time mean for northern hemisphere mid-high latitudes (as evidenced from in-situ measurements of the seasonal cycles of both alkanes and alkyl nitrates from Summit in Swanson et al. (2003).

However, extending this analysis to [OH]=$1.2 \times 10^6$ gives only slightly larger changes of 8.1% for Scenario A and 5.8% for Scenario B. So the point made, that these are very small compared to the factor of 4-5 changes seen in the alkyl nitrate scenarios, still holds.

A brief discussion of this is now included in the manuscript, with further discussion in Supplementary Information (see response to point above).

**P13 L28: Equation 13. [NO]/[HO2] should be presented with some sort of average symbol that is different from what you have now, which implies an instantaneous average of concentrations instead of a long term average. ie [NO]~ ∫ [NO]dt or [NO]/[HO2] = ∫ [NO]/[HO2] dt**

We agree, firstly we will replace the term $k_{14}[NO]/(k_{14}[NO]+k_{11}[HO_2])$ in Equations E4-E6 with the term $\gamma$, which we will define as the mean $k_{14}[NO]/(k_{14}[NO]+k_{11}[HO_2])$ to which the air masses are exposed during transport from source region to the Arctic. I.e.

$$1/t * \int k_{14}[NO]/(k_{14}[NO]+k_{11}[HO_2]).dt$$

We now do not now explicitly calculate changes to $[NO]/[HO_2]$ but when in places we do refer to this ratio we are careful to define precisely where these are relevant to / what we mean.

We have removed Figure 4, the plot of $\Delta[NO]/[HO_2]$, and we now have a plot of $\gamma$, i.e. the mean alkyl nitrate production efficiency, $1/t * \int k_{14}[NO]/(k_{14}[NO]+k_{11}[HO_2]).dt$. We agree that it is incorrect to then determine a mean $[NO]/[HO_2]$ ratio from a rearrangement of,

$$\gamma = \overline{\left(\frac{k_{14}[NO]}{k_{14}[NO] + k_{11}[HO_2]}\right)}$$

to something such as

$$\overline{\left(\frac{[NO]}{[HO_2]}\right)} = \frac{k_{14}}{k_{11}}\left(\frac{1}{\gamma} - 1\right)$$

Hence Figure 4 is now a plot of $\Delta\gamma$ relative to 1970. This has little effect on our conclusions and $\gamma$ then feeds in to Equation 6.

**E3: The [NO]/[HO2] average you calculate and present in Figure 4 depends (perhaps critically so) on the values of all rate constants and branching ratios in the equations. The constants assumed should be presented somewhere in the paper and/or you should tell us at what temperature you calculated the rate constants (since T likely changes 30-40 oC as the air mass is transported from source regions to the arctic the region). What is the uncertainty in assuming a constant T. What is sensitivity of Figure 4 to using different T?**

All rate constants and branching ratios are now given in a table which I have put in Supplementary Information, while commenting in the manuscript that all rate constants and branching ratios are taken from the MCMv3.3.1 for a temperature of 273K.

*"All rate constants and branching ratios used in the calculations are taken from MCMv3.3.1 (mcm.leeds.ac.uk) assuming a temperature of 273K."*

We assume a constant T of 273 K. Changing the temperature (but keeping it as a constant) has no effect on the relative change in $\gamma$ (i.e. Fig. 4 would be unchanged). Reducing the constant T by 15 K to 258 K, increases the calculated peak relative change in [OH].t (i.e. the results in Fig. 8) from a factor of 2.47 to 2.56 for 2-butyl nitrate, and from 2.89 to 3.05 for 2+3-pentyl nitrate (the [OH].t calculated from 3-methyl-2-butyl nitrate is unchanged because the iso-pentane+OH rate constant from MCMv3.3.1 is not T dependent). Increasing the constant T by 15 K to 288 K, decreases the [OH].t peak from 2.47 to 2.41 for 2-butyl nitrate and from 2.89 to 2.77 for 2+3-pentyl nitrate.

I cannot see that changing temperature could have had much of an effect, with an increase of ~1 K in NH mean winter time temperatures between the beginning and end of our record. Though temperatures may have changed a little more in the Arctic, processing is basically finished by the time the air mass reaches the Arctic because there is no sunlight and hence no photochemistry.

Changing source regions and hence transport time would likely also change the temperature profile to which the air mass was exposed but as shown above, changing temperature has a small effect on the calculations and the main effect would be the change to processing time rather than any change in temperature.

**Figure 14L4 (Figure 4 caption). Why is photochemical age ∫ [OH]dt = 5 x1011?? This is about 5 days averaging at global average [OH]. How does figure change as you change the photochemical processing?**

As described in the text:

*"This is based on a mean transport time of air masses from Europe (from where the majority of winter-time pollutants are transported to the Arctic – see Section 6.1) to the Arctic in the winter of ten days (Stohl, 2006), and a mean winter-time [OH] of ~ 6 × 10$^5$ cm$^{-3}$ (in reasonable agreement with that derived by Derwent et al. (2012) for the North Atlantic in winter-time)."*

We are not considering mean global [OH], we are considering a mean [OH] in mid-high latitude northern hemisphere winter-time (see comment above).

See discussion on next point for sensitivity analysis.

**p17E6 – I believe this equation is derived presuming that alkyl nitrates do not photolyze. Considering that ~ 50% of loss of 2-butyl nitrate is photolysis, what is the effect on the calculations and the results in Figure 6 resulting from this assumption.**

Yes, you are correct and we can (and should) include the photolysis sink. This has no effect on the conclusions of the paper. It changes the magnitude of the calculated **absolute** values for $\gamma$ but not the relative changes (i.e. Figure 4). For [OH]t it changes both the calculated absolute and relative changes.

For the sake of reducing the number of assumptions we have to make we include the photolysis sink, $j_{16}$, in the ratio, $\lambda$, where $\lambda = j_{16}/k_{15}\overline{[OH]}$, i.e. the ratio of the photolysis sink to the OH sink. Presenting it like this allows us to not have to assume a value for $j_{16}$ and means that [OH] drops out of the equation meaning we move from the equation below,

$$\gamma = \frac{[\text{RONO}_2](k_{15}[\text{OH}]+j_{16}-k_{13}[\text{OH}])}{[\text{RH}][\text{OH}]\beta k_{13}(1-e^{(k_A-k_B)t})} \tag{E5a}$$

to that now presented in the paper, Equation E5.

$$\gamma = \frac{[\text{RONO}_2](k_{15}(1+\lambda)-k_{13})}{[\text{RH}]\beta k_{13}(1-e^{(k_A-k_B)t})} \tag{E5}$$

In the Supplementary Information we show the sensitivity of the calculated relative changes to [OH]t to the assumed value of $\lambda$ (as stated above, relative changes to $\gamma$, and hence [NO]/[HO$_2$], are not affected by the inclusion of photolysis).

In Equation E5, $\lambda$ is assumed to be constant through the period of study (1960-2007) as [OH]t is constant. We assume a value of 1 for $\lambda$ in Figure 4 and look at the sensitivity of the calculated trend in $\gamma$ to this value.

In Equation E6, [OH] is still present on the right hand side of the equation in the term $\lambda$. A calculation is initially done assuming $\lambda$ to be constant through time and = 1. Applying the derived relative [OH] temporal trend obtained (as part of [OH]t) from this calculation to $\lambda$ gives a temporal trend in $\lambda$. The calculation is then redone applying this temporal trend and assuming that $\lambda$ in 1970 = 1. This process is repeated and we iteratively converge on a unique solution for the trend in [OH]t.

(All of this serves to reduce the calculated changes in [OH]t only slightly compared to the previous Figure 6).

The sensitivity of the calculated trend in [OH]t to various assumed values for $\lambda$ in 1970, and to assuming $\lambda$ to be a constant value, are shown in the Supplementary Information.

This is now described in the paper as,

"*Figure 7 shows the trends in $\overline{[OH]}t$ derived from the alkyl nitrate-alkane pairs if a constant value for $\gamma$ is assumed. The value used for the constant $\gamma$ for each alkyl nitrate was the mean value derived in Figure 4 for the period 1960 – 2007 (0.31 for 2-butyl nitrate, 0.34 for 2+3-pentyl nitrate, 0.17 for 3-methyl-2-butyl nitrate).*

*Equation E6 also has [OH] terms on the right hand side of the equation, incorporated in $\lambda$. The results in Figure 7 are determined through an iterative process of fitting a polynomial to the trend calculated using an a-priori assumption that $\lambda$ = 1 for the whole time period. The calculation is then redone with a temporally varying value for $\lambda$ using this fit to determine the changes. This converges towards the unique solution presented in Figure 7. E.g. In Figure 7 for 2-butyl nitrate, in 1970 the assumed value of $\lambda$ is 1, at the peak of $\Delta\overline{[OH]}t$ in 1997, when $\Delta\overline{[OH]}t$ = 2.1, the value of $\lambda$ is 0.46 (1/2.1).*"

**Also why is [NO]/[HO2] presumed to be 0.5. Reference? rationale? Sensitivity?**

Equation E6 is now presented as,

$$\overline{[OH]}t = ln\left(1 - \frac{[\text{RONO}_2](k_{15}(1 + \lambda) - k_{13})}{[\text{RH}]\gamma\beta k_{13}}\right) \div \left(k_{13} - k_{15}(1 + \lambda)\right)$$

The [NO] and [HO$_2$] are thus incorporated within $\gamma$. For each alkyl nitrate we then use the mean value of $\gamma$ between 1960-2007 calculated using Equation E5 (using a fixed value of [OH]t of $5\times10^{11}$ cm$^{-3}$ s, the rationale for this value is given in our answer two

points above – although as stated in the paper, the relative change in γ is actually unaffected by changing [OH]t).

In the Supplementary Information we show the sensitivity of the calculated relative changes to [OH]t to the assumed value of γ.

**Minor Points**

**p2L23-24: by removal, are you presuming CH4 +OH and CO + OH regenerate OH catalytically. In any case, please provide a reference for such statements.**

Yes, we are talking about primary sources of OH. However, you are right we should provide more detail. For both this point and the next, what we are trying to do in the Introduction is to provide a succinct summary of tropospheric $NO_X$-$HO_X$-$O_3$ chemistry, which I'm sure you appreciate is something of a challenge.

We have added the following text to the Introduction,

*"Other reactions, such as alkene ozonolysis (Johnson and Marston, 2008) and photolysis of HONO (formed from heterogeneous reactions of $NO_2$ (Stone et al. 2012)) may also be important primary sources of $HO_X$, particularly in winter (e.g. Heard et al., 2004)."*

*"The main removal process for HOx in urban regions is the reaction of OH with $NO_2$ (Reaction R5) (Stone et al., 2012) while $HO_2$ self-reaction and reaction with $RO_2$ (in particular $CH_3O_2$ + $HO_2$) (Reactions R11-R12) dominate in low NOx environments (Stone et al., 2012)."*

**page 2 - HOx Sources: In winter regions, O3 photolysis may not be the main source of OH. There is lots of recent evidence for this.**

Yes, we should mention this is in the introduction (we do mention alkene ozonolysis as a major source of OH Pg21, line 21.) Additionally we will mention that HONO, may be a primary source of OH, although there are still very large uncertainties in the role of HONO and much of it also likely comes from OH+NO.

See point above for text added to the Introduction.

**Page 2-3 and throughout: The chemical equations are not presented with care. Three body reactions should be presented as such (R2 for example). Photolysis reactions should be indicated as photolysis reactions. (R1, R3, etc). R7 is an equilibrium reaction. Reaction 13 is not balanced, namely because it is actually 2 reactions. R14a does not exclusively give aldehydes as shown, it also give ketones (ie- MEK from butane). R14b involves a rearrangement that is apparently highly temperature dependent that is not mentioned.**

Apologies for this, it was the result of having written and re-written the paper numerous times. These will be tidied up and corrected as suggested.

**p4L26 – Emissions definitely did have seasonal patterns in the past. I am not sure about today.**

See discussion on Major Concern.

**p7L27 – Your answer to the question by a reviewer about the effect of the 34.7m sample was not convincing. If this sample was removed, do your conclusions change; regardless of whether it should be removed or not. The historical state of the atmosphere should not rest on a single point.**

No, our conclusions would not change.

Firstly, we primarily focus on the stark difference between alkyl nitrate mixing ratios in the 60s-70s and those in the mid-1990s. This is a trend that is seen very clearly in the records of all three alkyl nitrates presented here and is also seen in trend of the same three alkyl nitrates taken from the NGRIP site in Greenland seven years earlier, presented in Worton et al. (2010).

Removing the measurement at 34.7 m has no effect on this.

Secondly, if this measurement is removed, the scenario for 2-butyl nitrate remains rather similar (falling to ~4.2 ppt by the end of the record rather than 3.5 ppt), 3-methyl-2-butyl nitrate falls to ~1.2 ppt rather than 0.8 ppt. It is only 2+3-pentyl nitrate for which the post 1995 part of the reconstruction would change significantly, remaining almost flat rather than decreasing as for the other two.

The pentyl nitrate measurements have a greater scatter in the peak region than the butyl nitrate measurements. This is reflected in the larger 2-sigma uncertainty envelopes of the atmospheric reconstructions.

**Figure 3 – how many points are represented in the curve for n-butane in this figure. How uncertain is the peak year, or put another way, what is the uncertainty in the difference between the peak year for n-butane and 2-butyl nitrate. I presume that the shape of Figures 4 and Figures 6 depend critically on the relative temporal trend of n-butane measured in the firn from the other paper. Does the shape of n- butane in Figure 3 make sense given the temporal changes in hemispheric emissions of n-butane...can it be corroborated with other sources of hydrocarbon measurements in northern hemisphere cities or other sites??**

The output is from the firn model. Hence the output frequency can be set at whatever we want. The output frequency for the curve shown is monthly. Of course the firn model output represents a smoothed temporal average and cannot represent seasonal or short scale inter-annual variations, particularly in the earlier part of the

record. It represents the temporal trend in surface atmospheric concentrations above the firn that best reproduces the measurement depth profile.

The peak timing of all of the atmospheric scenarios derived for the C3-C5 alkanes (propane - 1979, n-butane - 1980, iso-butane - 1980, n-pentane - 1981, iso-pentane - 1981) measured at NEEM in Helmig et al. (2014) agree to within two years. They also agree to within five years (propane - 1980, n-butane - 1977, iso-butane - 1977, n-pentane - 1976, iso-pentane - 1978) with the peaks derived in Worton et al. (2012) measured at a different Arctic site, North GRIP.

Emission estimates for the butanes and pentanes are available from the ACCMIP data set, available from the ECCAD website (eccad.sedoo.fr). These suggest very similar emissions trends for the regions Europe and North America with emissions rising steadily from the 1950s to 1980, before declining to about 1965 levels 2000. This trend in alkane emissions is broadly consistent with the atmospheric concentrations derived from the firn air.

**p11L4. At least once in this paper you should acknowledge that [OH]t is not a constant, it represents ∫ [OH] dt, and perhaps it would better be presented as an average symbol.**

Lines 6 - 8 in the new version of Section 4 now reads,

*"Similarly for the purposes of this work, [OH] is assumed to represent an average [OH] to which the air mass is exposed during transport from the source region to the Arctic, i.e. 1/t * ∫[OH].dt."*

We now present [OH] when used as $\overline{[OH]}$.

**p13L15 – [NO] can range from 1ppb to ??? I have never seen the upper end of your range, 1000 ppb?? Provide a reference if so.**

The DEFRA Air Quality Expert Group 2004 report "Nitrogen Dioxide in the United Kingdom" available from http://www.defra.gov.uk/environment/airquality/aqeg gives NOx concentrations that range typically up to a few hundred $\mu gm^{-3}$ ($\mu gm^{-3}$ ~ 2*ppb for $NO_2$) and in extreme cases going up to 3000 $\mu gm^{-3}$ (i.e. ~1.5 ppm).

However, since we refer to [NO], we have changed this line to,

*"In an urban environment, daytime [NO] can range from ten to a few hundred ppb."*

**P15 Figure 5 –for comparison, both y-axis should extended to zero. Units missing on right axis.**

The units are now included on the right hand axis and both axes extend to zero.

**p16L20 Section 5.2. Does 1 sentence deserve its own section?**

No, perhaps not. We have moved this line to the end of Section 5.0.

**p20L12-15: other primary sources of O3. HCHO? HONO??**

I'm not sure what you are referring to with this page and line reference. I'm guessing you mean other primary sources of OH? If so, we will extend the discussion here in a similar way to in the introduction.

*"Another primary OH source is via the ozonolysis of alkenes (Johnson and Marston, 2008). A third source of OH that may be important is photolysis of HONO (e.g. Stone et al., 2012). There is still considerable uncertainty about the sources of HONO, with formation from heterogeneous conversion of $NO_2$ via a range of postulated processes appearing to dominate over the $HO_X$ / $NO_X$ recycling reaction OH + NO (e.g. Michoud et al., 2014). This again would be a primary source of OH which would be expected to correlate positively with $NO_X$ concentrations."*